# Mutant p53 drives an immune cold tumor immune microenvironment in oral squamous cell carcinoma

Yewen Shi [1,2], Tongxin Xie[1], Bingbing Wang[1], Rong Wang[3], Yu Cai[4], Bo Yuan[5], Frederico O. Gleber-Netto [1], Xiangjun Tian[6], Alanis E. Rodriguez-Rosario[1,7], Abdullah A. Osman[1], Jing Wang [6], Curtis R. Pickering[1], Xiaoyong Ren[2], Andrew G. Sikora[1], Jeffrey N. Myers [1✉] & Roberto Rangel [1✉]

The critical role of the tumor immune microenvironment (TIME) in determining response to immune checkpoint inhibitor (ICI) therapy underscores the importance of understanding cancer cell–intrinsic mechanisms driving immune-excluded ("cold") TIMEs. One such cold tumor is oral cavity squamous cell carcinoma (OSCC), a tobacco-associated cancer with mutations in the *TP53* gene which responds poorly to ICI therapy. Because altered *TP53* function promotes tumor progression and plays a potential role in TIME modulation, here we developed a syngeneic OSCC models with defined *Trp53* (*p53*) mutations and characterized their TIMEs and degree of ICI responsiveness. We observed that a carcinogen-induced *p53* mutation promoted a cold TIME enriched with immunosuppressive M2 macrophages highly resistant to ICI therapy. *p53*-mutated cold tumors failed to respond to combination ICI treatment; however, the combination of a programmed cell death protein 1 (PD-1) inhibitor and stimulator of interferon genes (STING) agonist restored responsiveness. These syngeneic OSCC models can be used to gain insights into tumor cell–intrinsic drivers of immune resistance and to develop effective immunotherapeutic approaches for OSCC and other ICI-resistant solid tumors.

[1] Department of Head and Neck Surgery, The University of Texas MD Anderson Cancer Center, Houston, TX 7030, USA. [2] Department of Otorhinolaryngology Head and Neck Surgery, The Second Affiliated Hospital of Xi'an Jiaotong University, Xi'an 710004, China. [3] Department of Endodontics, School & Hospital of Stomatology, Wuhan University, Wuhan, China. [4] Department of Oral & Maxillofacial Surgery, School & Hospital of Stomatology, Wuhan University, Wuhan, China. [5] Department of Pulmonary Medicine, The University of Texas MD Anderson Cancer Center, Houston, TX 77030, USA. [6] Department of Bioinformatics and Computational Biology, The University of Texas MD Anderson Cancer Center, Houston TX 77030, USA. [7] Department of Biology, University of Puerto Rico, Bayamon, Puerto Rico, USA. ✉email: jmyers@mdanderson.org; rrangel@mdanderson.org

Head and neck squamous cell carcinoma (HNSCC), the sixth most common cancer worldwide, results in more than 350,000 deaths every year[1]. Most HNSCCs are oral cavity squamous cell carcinomas (OSCCs), more than 90% of which arise from premalignant precursor lesions[2–4]. In 2020, 377,713 cases of oral cavity and lip cancer were reported, and 177,757 patients died of this disease[5]. Although many locally advanced OSCCs respond initially to the multi-modality treatment with surgery, radiation therapy, and chemotherapy, patients remain at high risk of post-treatment and/or distant metastasis. Cytotoxic chemotherapy has limited efficacy against metastatic OSCC, and patients with recurrent/metastatic disease have a median overall survival duration of less than 1 year[6].

Immune checkpoint inhibitors (ICIs), such as antibodies against programmed cell death protein 1 (PD-1) and programmed death ligand 1 (PD-L1), have revolutionized the treatment of many cancers, including OSCC[7]. However, the overall response rate of HNSCC patients receiving PD-1 inhibitors is less than 20%, regardless of human papillomavirus (HPV) infection status[8,9]. The progressive acquisition of genetic alterations in epithelial cells during oral cancer development, leading to unfavorable changes in the tumor immune microenvironment (TIME), plays a major role in OSCC resistance to therapy, including immunotherapy[10]. Many OSCC are considered poorly immunogenic tumors or "immune deserts" which lack immune infiltration, evade immune recognition and suppress immune system activation, all of which have been associated with early disease relapse and poor prognosis in OSCC/HNSCC patients[11,12]. However, as yet few specific mechanisms by which cancer-specific mutations modulate the TIME, and thereby impact ICI response, have been described.

Somatic TP53 mutations, the most common genetic alterations across all cancers[13], occur in 75–85% of non–HPV-associated HNSCCs, including OSCCs. Although some TP53 mutations lead to a loss of wild-type (WT) p53 function, many TP53 mutations confer gain-of-function (GOF) activity that promotes invasion, metastasis, genomic instability, proliferation, and tumor-associated inflammation[14]. The loss or mutation of TP53 in a cancer can affect the recruitment and activity of myeloid cells and T cells, thereby enabling immune evasion and tumor progression[15]. Moreover, mutant TP53 can modulate the TIME by regulating proinflammatory cytokine signaling pathways, which can inactivate the innate immune response by altering signaling through the cyclic GMP-AMP synthase (cGAS)–stimulator of interferon genes (STING) pathway, thus reducing the infiltration of cytotoxic $CD8^+$ T cells[16–19]. Overall, genomic alterations in TP53 contribute to tumorigenesis by driving the growth and survival of the epithelial tumor compartment and by enabling immune evasion in the TIME of various cancer types[15]. However, specific mechanisms by which mutant TP53 modulates the tumor microenvironment (TME) in OSCC have yet to be described.

To better understand the role that expression of mutant p53 in epithelial cells has in shaping the TIME, we generated a set of syngeneic mouse oral cancer cell lines (ROCs) with p53 mutations, which are different from the oligodendrocyte cell line Roc-1[20]. We developed these cell lines from either genetically engineered mouse models or WT mice with Trp53 (p53) mutations acquired by exposure to 4-nitroquinoline-1 oxide (4-NQO), a carcinogen that acts as a tobacco mimetic and causes DNA damage. All three of the syngeneic murine ROC cell lines were tumorigenic and had different tumor molecular characteristics and immune landscapes. We used the ROC1 cell line to investigate the effect of mutant p53 in the modulation of cell-intrinsic factors that shape the tumor immune landscape and affect sensitivity to immunotherapy. Our syngeneic OSCC models provide an experimental system which can be used to understand the interplay between cell-intrinsic genetic changes and immunosuppressive mechanisms that promote tumor progression, and serve as a translationally relevant platform for evaluating immunotherapy combinations to improve treatment strategies for OSCC.

## Results

**Syngeneic mouse oral cancer cell lines are faithful to the mutation and expression landscape signatures of human OSCC.** The mouse model of 4-NQO–induced chemical carcinogenesis is a representative OSCC model since it recapitulates the sequential stages of oral carcinogenesis observed clinically in OSCC; in addition, 4-NQO is a DNA adduct–forming carcinogen that acts as a tobacco mimetic and causes genomic alterations similar to those in human OSCC[21].

We provided 15 transgenic mice with drinking water containing 100 μg/mL 4-NQO for 8 wk; at 22–24 wk, tongue tumors were harvested and sectioned for histological processing and tissue dissociation to establish C57BL/6 syngeneic ROC oral cancer cell lines. Histological tumor slides were reviewed independently by two pathologists who made the diagnosis of OSCC (Supplementary Fig. 1b). The three cell lines that we developed were designated ROC1 through ROC3 (Fig. 1a). The ROC1 cell line was derived from p53 WT mice (K14 Cre$^{Tg/+}$; p53$^{wt/wt}$); the ROC2 line was derived from a p53 loss of function (LOF) mouse (K14 Cre$^{Tg/+}$; p53$^{flox/flox}$); and the ROC3 line was derived from a p53 R172H transgenic mouse (K14 Cre$^{Tg/+}$; p53$^{R172H/flox}$).

First, we sorted high EGFR$^+$ tumor cells by fluorescence-activated cell sorting to eliminate fibroblasts and enrich for tumorigenic cells (Supplementary Fig. 1c). Next, the ROC1, 2, and 3 cell lines were cultured and selected for their capacity to generate cancer colonies in soft agar. After several weeks in culture, the soft agar colonies were selected and plated in tissue culture to finally establish the tumorigenic ROC cell lines (ROC1–3) (Supplementary Fig. 1c).

Sanger DNA sequencing confirmed the p53 R172H mutation in the ROC3 cell line. Moreover, we identified a p53 T122N missense mutation in the ROC1 cell line, consistent with the known occurrence of TP53 mutations as an early event in development human oral cancer. Interestingly, p53 mutation in the ROC1 cell line showed p53 amino acid mutation positions similar to those in MOC22 and MOC-L1 oral cancer cell lines[22,23]. Although the ROC and MOC22 cell lines were generated with different carcinogens, this further emphasizes the role of altered p53 as a trunk genetic driver of OSCC. Moreover, we characterized the three ROC cell lines with use of colony formation, cellular proliferation, wound healing, and invasion assays (Supplementary Fig. 1d–g). ROC3 had higher rates of clone formation and proliferation, whereas ROC1 and ROC2 had higher rates of cell migration and invasion.

Next, we performed whole-exome sequencing (WES) of the three ROC lines. The WES reads were mapped against the reference genome of GRCm38 of Mus musculus strain C57BL/6J by using Burrows-Wheeler Aligner (BWA) (v0.7.17), and the single-nucleotide variant (SNV) and insertion/deletions (INDELs) were called by using VarScan2 (min-coverage = 10, P = 0.01). The coverage depth of the whole exome ranged from 637X to 670X (mean = 659X) (Supplementary Table 1). Analysis using the VarScan platform detected approximately 4000 single-nucleotide variants in each of the ROC1, and ROC2 cell lines; interestingly, the ROC3 cell line contained about 10,000 single-nucleotide variants, of which nearly 40% were exonic mutations (Supplementary Table 2). Furthermore, we found that C>T|G>A, and A > G|T > C mutations were dominant in ROC1, and ROC2, whereas C > A|G > T mutations were dominant in ROC3. Remarkably, C > A|C>T mutations were tobacco mutation signatures, confirming that 4NQO closely mimics the effects of tobacco in human oral cancer (Fig. 1b)[24].

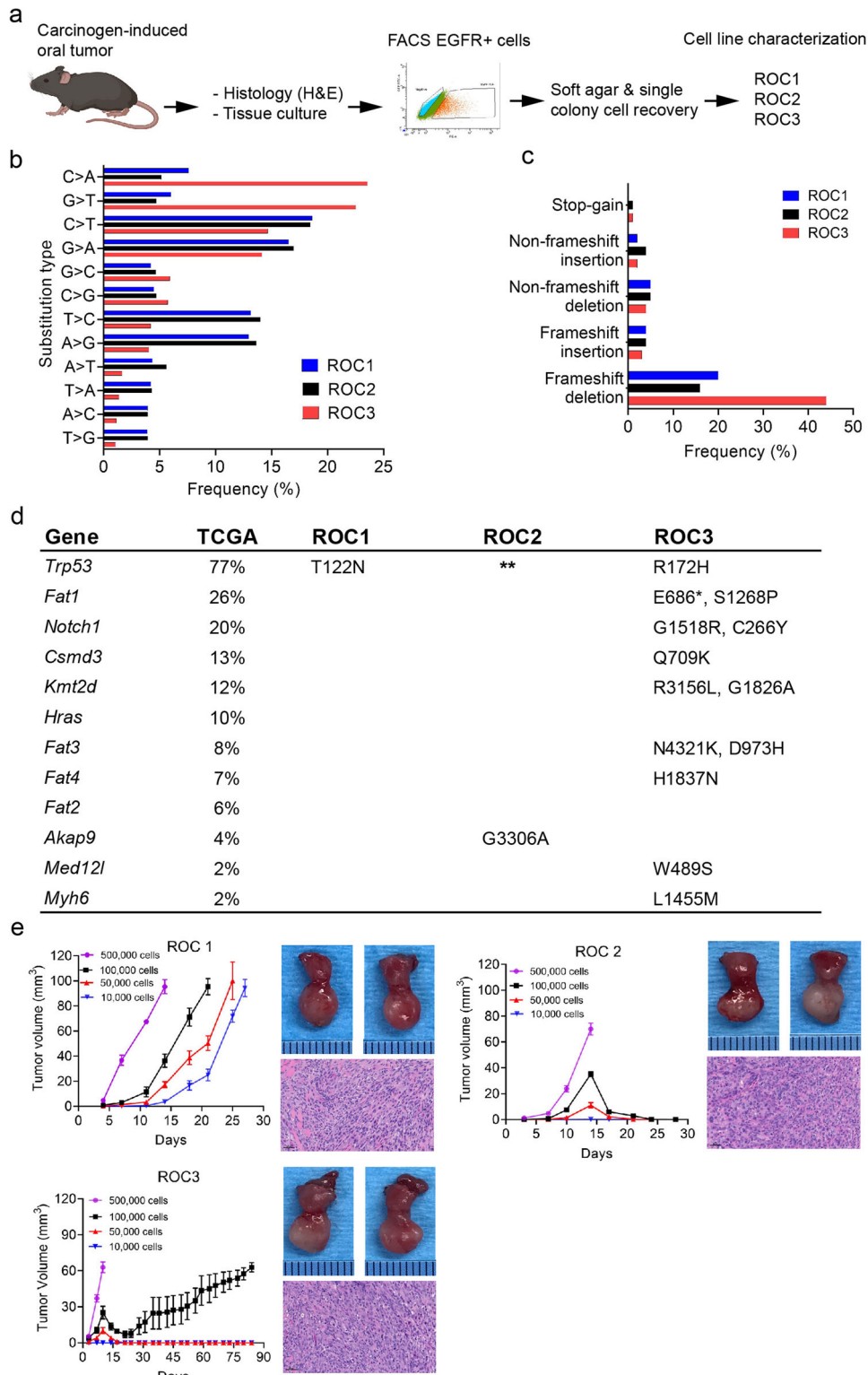

**Fig. 1 Mutational landscape and tumorigenesis of syngeneic ROC cell lines derived from a murine model of carcinogen-induced oral cancer. a** Strategy for generating syngeneic ROC mouse oral cancer cell lines. **b** Frequency of somatic substitutions in ROC cell lines. **c** Frequency of functional INDELs in ROC cell lines. **d** Comparative analysis among ROC cell lines of the genes most frequently mutated in the TCGA-HNSCC cohort. *p53 nonsense mutation; **p53 deletion. **e** Tumor growth curves of different numbers of injected ROC1–3 cells in the orthotopic tongue model. Mean tumor volume are represented in the graph ($n = 5$, error bars = standard error mean). The right panels show the primary tumor in the tongue (top) and a representative hematoxylin and eosin–stained section of the primary tumor (bottom). Scale bars, 50 μm.

In each ROC cell line, we detected about 1000 INDELs, 76–82% of which were at intronic regions and 1.7–5.3% of which were exonic mutations (Supplementary Table 3). Moreover, the ROC3-showed the highest INDEL frequency in ROC cell lines (Fig. 1c). Furthermore, we assessed the association between the ROC cell line mutations and the most significantly mutated genes in The Cancer Genome Atlas (TCGA)-HNSCC cohort; the mutation frequency was based on 285 HPV- HNSCC tumors from the oral cavity site. The data were from the TCGA PanCanAtlas PanSquamous project[25] and cBioPortal.org[26,27]. Bioinformatic studies detected acquired mutations in *p53*, *Fat1*, *Notch1*, *Csmd3*, *Kmt2d*, *Fat3*, *Fat4*, *Akap9*, *Med12l*, and *Myh6*, in murine ROC cell lines (Fig. 1d), as reported previously in other murine oral cancer lines and human HNSCC[28–33]. In addition, we included the allele frequency and depth of the mutated genes in the ROC cell lines (Supplementary Table 4). Overall, these findings show that the murine ROC cell lines are a relevant preclinical syngeneic oral tumor model with which to study OSCC.

To model oral cancer, we orthotopically transplanted various concentrations of ROC1–3 cell lines into the tongues of immunocompetent C57BL/6 mice[34]. These mice were then given orthotopic injections of 500,000 cells/mouse into the oral tongue, and oral tumors developed from all three cell lines, as indicated by hematoxylin and eosin staining (Fig. 1e). Even with low-concentration injections (10,000 cells), mice given cells from ROC1 developed tumors, and ROC3 generated tumors after injections of at least 100,000 cells (Fig. 1e). These results indicate that the ROC cell lines with germline and carcinogen-acquired *p53* mutations have different tumorigenicity levels.

To investigate additional similarities to human OSCC, we analyzed the capacity of orthotopic tumors to metastasize to cervical lymph nodes. Histological evaluation of cervical lymph nodes confirmed that only ROC1, and ROC3 tumors metastasized to these nodes. Orthotopic injections of 500,000, 100,000, 50,000, and 10,000 ROC1 cells resulted in cervical lymph node metastasis rates of 80%, 40%, 40%, and 60%, respectively. Injections of 500,000 ROC3 cells yielded a cervical lymph node metastasis rate of 60%.

Further IHC analysis of mesenchymal and epithelial markers revealed low levels of keratin-14 (K14) and high levels of vimentin expressed in ROC1 and ROC2 tumors; high levels of K14 and low levels of vimentin were expressed in ROC3 tumors. Ki67 staining revealed that all of the ROC tumors had a high proliferation rate (Supplementary Fig. 2c). Opal multiplex IHC detected different p53 protein levels and patterns of subcellular distribution in the ROC tumors. Of interest, ROC3 tumors showed a nuclear distribution of p53, whereas ROC1 tumors had lower levels of p53 in the cytoplasm and nucleus (Supplementary Fig. 3).

**ROC1 tumors have cold TIME landscapes that are resistant to immunotherapy.** ROC2 and ROC3 tumors had a higher infiltration of CD8[+], CD4[+], and Foxp3[+] T cells in the TIME than ROC1 tumors (Fig. 2a). Only ROC3 tumors had a high infiltration of CD11c[+] cells. Although all ROC tumors expressed tumor-associated macrophage markers (e.g., CD68), only ROC1 and ROC3 tumors expressed more CD206, a marker of tumor-associated M2 macrophages (Fig. 2b). Furthermore, ROC2 and ROC3 tumors had abundant expression of immune checkpoints PD-1 and TIGIT (Fig. 2c). These results indicate that ROC1 tumors are cold tumors that lack immune effector cells, whereas ROC2 and ROC3 tumors are warm tumors with a high infiltration of effector and immunosuppressor immune cells expressing different immune checkpoints to evade immunosurveillance.

The ROC1 cell line, with its spontaneous p53 T122N mutation, gave rise to tumors with an aggressive phenotype similar to that of human OSCC, which included high tumorigenicity, a high rate of lymphatic metastasis, and an absence of CD8[+] T-cell infiltration. Therefore, we selected ROC1 tumors to interrogate whether ICI could cause tumor regression (Fig. 3a). ROC1 tumors failed to respond to treatment with antibodies against PD-1 and TIGIT (Fig. 3b, c), confirming that ROC1 is a cold tumor lacking immune cell infiltration and immune checkpoint expression (Fig. 2 and Supplementary Fig. 4) and thus a good model of ICI-resistant disease. Because we used mouse-specific antibodies against PD-1 or TIGIT, treatment failure was not likely due to development of neutralizing antibodies against human IgG.

**Mutant p53 modulates the immunosuppression mechanisms of ROC1 cold tumors.** Because only 10–15% of patients with advanced-stage HNSCC (including OSCC) respond to checkpoint inhibition, we investigated the immunosuppression mechanisms of ROC1 tumors. As described above, ROC1 cells acquired a *p53* mutation; such an alteration occurs in 65% to 85% of OSCC patients[35,36], suggesting that *p53* mutations leading to altered TIME are highly clinically relevant. ROC1 cells stably expressing NTC shRNA (control) or p53-shRNA knockdown (p53-KD) were generated (Fig. 4a) and orthotopically injected into C57BL/6J mice. Only mice injected with control or parental ROC1 cells developed tumors.

Next, to determine whether the immune system or tumor cell-intrinsic factors mediate tumorigenesis, we orthotopically injected ROC1 cells into immunodeficient C57BL/6J beige mice. Parental, control, and p53-KD ROC1 cells all formed tumors, suggesting that mutant p53 modulates immunosuppression mechanisms in these mice (Fig. 4b). However, the growth of the p53-KD ROC1 tumors was slightly delayed in beige mice (Fig. 4c), possibly owing to tumor cell-intrinsic factors necessary to stimulate stromal cell–tumor cell interactions.

We next performed RNAseq of ROC1 tumors grown in immunocompetent C57BL/6J mice to assess immune infiltration and identify tumor cell-intrinsic factors that might promote tumorigenesis and immunosuppression mechanisms. Bioinformatics analysis revealed that ROC1 tumors expressed markers of macrophages and Tregs as well as immune checkpoints associated with these cell types (Supplementary Fig. 5a).These results support previous IHC findings showing that Tregs and macrophages are known to promote and sustain a cold TIME[37,38]. Interestingly, we found upregulation of galectin-9 (Lgals9), which supports the polarization of CD206[+] macrophages (which are similar to the cells detected in ROC1 tumors) to support tumor growth[39]. We also detected high expression levels of proinflammatory cytokines and chemokines in the TME (Supplementary Fig. 5b). Notably, ligand/receptor pairs such as Il33/Il1rl1 (St2) and Cxcl12/Cxcr4 not only were abundant in tumors but also detected in cultured tumor cells. Such tumor cell-intrinsic factors are known to mediate an immunosuppressive microenvironment in different tumor types[40,41].

Because dysregulated cytokine and chemokine signaling in the TME favors tumor growth, the exclusion of effector immune cells, and the accumulation of immunosuppressive cells, we hypothesized that mutant *p53* T122N regulates the expression and secretion of these factors. We assessed the transcriptome analysis in ROC1 p53-KD compared to control cells and identified a dramatic enrichment of proinflammatory cytokines and chemokines confirmed by differential expression gene analysis (Fig. 4d). Interestingly, the top four pathways altered were the TNF, IL17, chemokine, and cytokine–cytokine receptor interaction pathways (Supplementary Data File 1), which suggests that mutant *p53* modulates the expression and secretion of soluble factors in the TME. Interestingly, our in silico analysis correlated with the

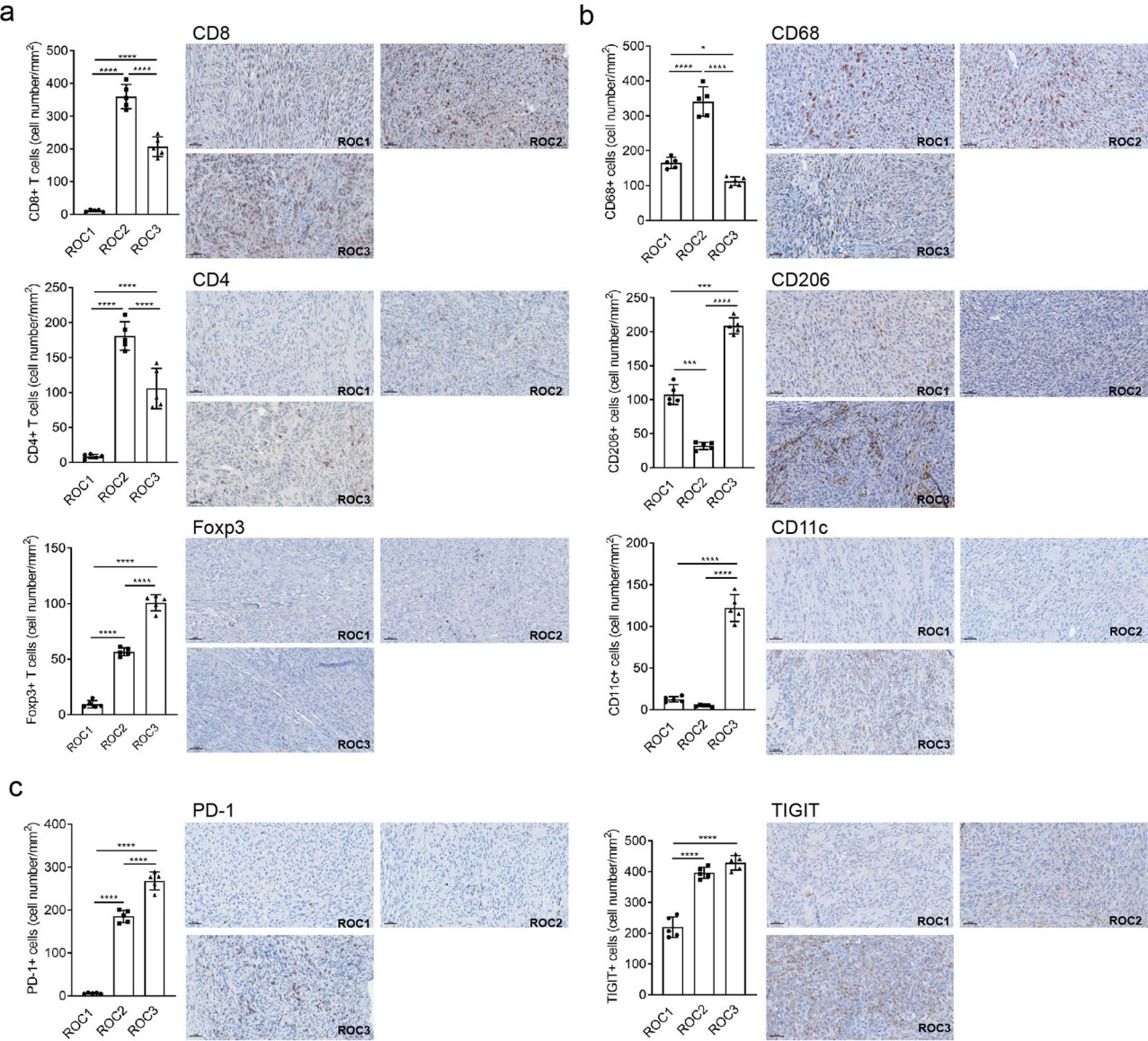

**Fig. 2 ROC1–3 TIMEs have distinct patterns of immune infiltration and exclusion.** IHC quantification analysis shows that T-lymphocyte markers (**a**), macrophage markers (**b**), and immune checkpoints (**c**) are differentially expressed among ROC1–3 tumors. The images were scan using Vectra Polaris imaging system and images were process using Phenochart software (version 1.0.12). Representative antibody-stained images are shown and include scale bars 50 μm (bar representative of $n = 3$ stained tumors, error bars= standard deviations). One-way ANOVA with Tukey's post hoc test $p$-values shown, $*p < 0.05$, $***p < 0.005$, $****p < 0.0001$, for all comparisons of the ROC tumors (error bars = standard deviations).

qPCR validation of the chemokines and cytokines that were upregulated (*Ccl2*, *Cxcl5*, *Ccl5*, *Cxcl16*, *Il6*) and downregulated (*Il17c*, *Il33*, *St2*) in the p53-KD compared to control ROC1-tumor cells (Fig. 4e). These results strongly suggest that mutant *p53* modulates the secretome of ROC1-tumor cells to influence the TME.

Given the dysregulated gene expression of cytokines and chemokines in the ROC1 tumors, we examined and validated the protein secretion levels of cytokines, chemokines, and other secreted factors that contribute to a cold TIME. We prepared conditioned media from cultured control and p53-KD ROC1-tumor cells for mouse cytokine arrays. In agreement with our gene expression studies, p53-KD ROC1-tumor cells showed an upregulation and secretion of cytokines and chemokines, including IL-10, C-C motif chemokine ligand 2 (CCL2), CCL5, CCL20, C-X-C motif chemokine ligand 9 (CXCL9), CXCL12, CXCL16, and pro-matrix metallopeptidase 9 (MMP9) (Fig. 4f),

soluble factors that promote the infiltration and activation of CD8$^+$ T cells and natural killer (NK) cells in tumors[42–47]. In ROC1 p53-KD cells, qPCR studies revealed that IL33 and ST2 receptor were both downregulated (Fig. 4e); the IL33/ST2 signaling pathway has been reported to sustain tumor-associated macrophages and activate tumor growth factor-beta (TGF-β) expression to sustain a signaling loop to promote cancer progression[48]. These data demonstrate that mutant p53 coordinates the expression of cytokines and chemokines to sustain tumor growth but inhibits the production and secretion of factors that attract cytotoxic T and NK cells[42–47]. We also found that ROC1 p53-KD cells abundantly secreted the pleotropic cytokine IL-6, which can modulate immune cell polarization depending on other cytokines expressed in the TME (Fig. 4f).

On the basis of these findings, we propose a molecular model in which mutant *p53* T122N modulates the expression and secretion of cytokines, chemokines, and tumor growth factors

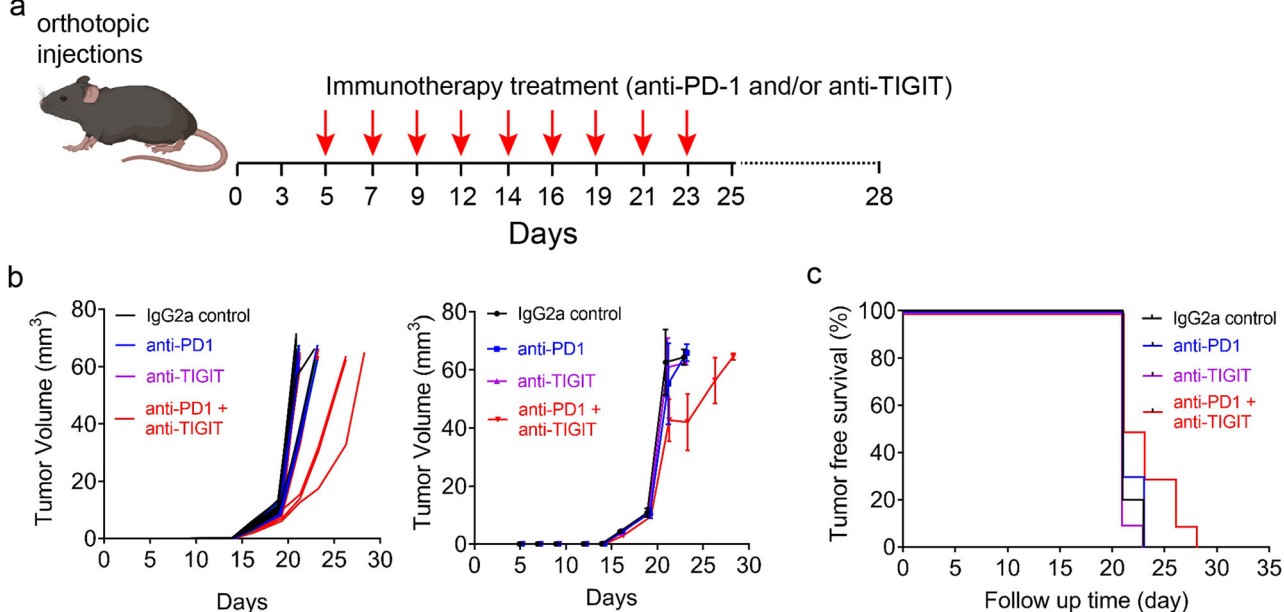

**Fig. 3 Immune-excluded mutant p53–mediated ROC1 tumors do not respond to immune checkpoint inhibition. a** Process diagrams of tumor cells orthotopically injected and treated with anti–PD-1 and/or anti-TIGIT antibodies (50,000 cells implanted; $n = 10$ mice). **b** ROC1-tumor-bearing mice that received the combination therapy had a slight robust tumor response. Left and right graphs represent individual and mean tumor growth (error bars = standard deviation). **c** Tumor-free survival with not significant overall survival (Long-Rank/Mantel–Cox test).

that support a cold TIME by polarizing M2 macrophages and recruiting immunosuppressive Tregs to promote tumor progression in immunocompetent mice (Fig. 4g).

**Mutant p53 modulates T-cell and macrophage polarization in cold OSCCs.** We next investigated the role of tumor-derived factors in the polarization of macrophages and the differentiation of T cells. We exposed bone marrow cells (BMCs) and splenocytes to ROC1 control– and ROC p53-KD–conditioned media for 72 h and identified the various immune cell populations with use of flow cytometry (Supplementary Figs. 6, 7)[49]. ROC1 p53-KD–conditioned media promoted the differentiation of T lymphocytes into CD8+ IFNγ+ and CD4+ IFNγ+ T cells (Fig. 5a, b) and decreased T-cell polarization into Treg (CD4+ Foxp3+) (Fig. 5c). BMCs exposed to ROC1 p53-KD–conditioned media facilitated the polarization of macrophages into the M1 phenotype, evidenced by the upregulation of M1 markers and the down-regulation of the M2 markers (Fig. 5d). These results might explain the lack of tumor development in ROC1-tumor cells lacking mutant p53 expression in immunocompetent mice. The ROC1 control- and ROC1 p53-KD-conditioned media had no effect on other immune cell populations; for example, the media did not affect the percentages of CD8+ and CD4+ in total T cells, Th2 cells (CD4+ IL-4+), Th17 cells (CD4+ IL-17+), exhausted T cells (CD3+ PD-1+), neutrophils, granulocyte-myeloid-derived suppressor cells (G-MDSCs), or and monocyte myeloid-derived suppressor cells (M-MDSCs) (Supplementary Fig. 8a–f). These data confirmed that secreted factors modulated by mutant p53 have a profound impact on the TIME.

**ROC1 cold tumors respond to combined immunotherapy.** Given that ROC1 cold tumors lack the infiltration of antigen-specific effector T cells, we next assessed whether a STING agonist could convert a cold tumor to a warm tumor and thereby improve the efficacy of an anti-PD-1 ICI[50,51]. Our immunotherapy studies showed that ROC1 tumors do not respond to anti-TIGIT or anti-

PD-1 ICIs alone or in combination (Fig. 3b, c). We hypothesized that the activation of STING by the intratumoral administration of a STING agonist (c-di-GMP) could stimulate the secretion of interferons and other cytokines that promote the antitumor response. ROC1-tumor cells were injected into the tongues of immunocompetent C57BL/6 mice, which were randomized to receive PBS, the STING agonist, a PBS/IgG2a isotype control, or anti–PD-1 antibody plus the STING agonist. Treatment was initiated when tumors reached a diameter of 2–3 mm (Fig. 6a, b). The intratumoral delivery of the STING agonist inhibited tumor growth in and prolonged the survival of ROC1 tumor-bearing mice (Fig. 6c, d). The combination of c-di-GMP and anti-PD-1 antibody was associated with a significant reduction in tumor burden and prolonged survival (Fig. 6e, f).

To further investigate the influence of the STING agonist on the TIME of ROC1 cold tumors, we harvested ROC1-tumor tissues from mice in the PBS control and c-di-GMP treatment groups at the indicated times (Fig. 6c; orange arrows) and found that STING expression was increased after the intratumoral delivery of c-di-GMP (Fig. 7a). Compared with those in the PBS control group, ROC1 tumors in the c-di-GMP group had greater infiltration of CD8+ and CD4+ T cells. In addition, c-di-GMP–treated tumors had high expression of granzyme B, a marker of activated CD8+ T cells, whereas control tumors had almost no expression of the marker (Fig. 7b). The STING agonist also inhibited the differentiation of CD4+ T lymphocytes into CD4+ Foxp3+ Tregs (Fig. 7b). Interestingly, although the STING agonist promoted the infiltration of macrophages, the percentage of M2 macrophages was reduced (Fig. 7c). Moreover, with the increase in CD8+ T cells in the TIME, the number of CD8+ PD-1+ cells was augmented in the c-di-GMP therapy group (Fig. 7d).

**Discussion**
We generated a panel of C57BL/6 syngeneic tumor cells that mimic human OSCC. These murine ROC cell lines have genomic alterations similar to those found in tobacco-associated OSCC;

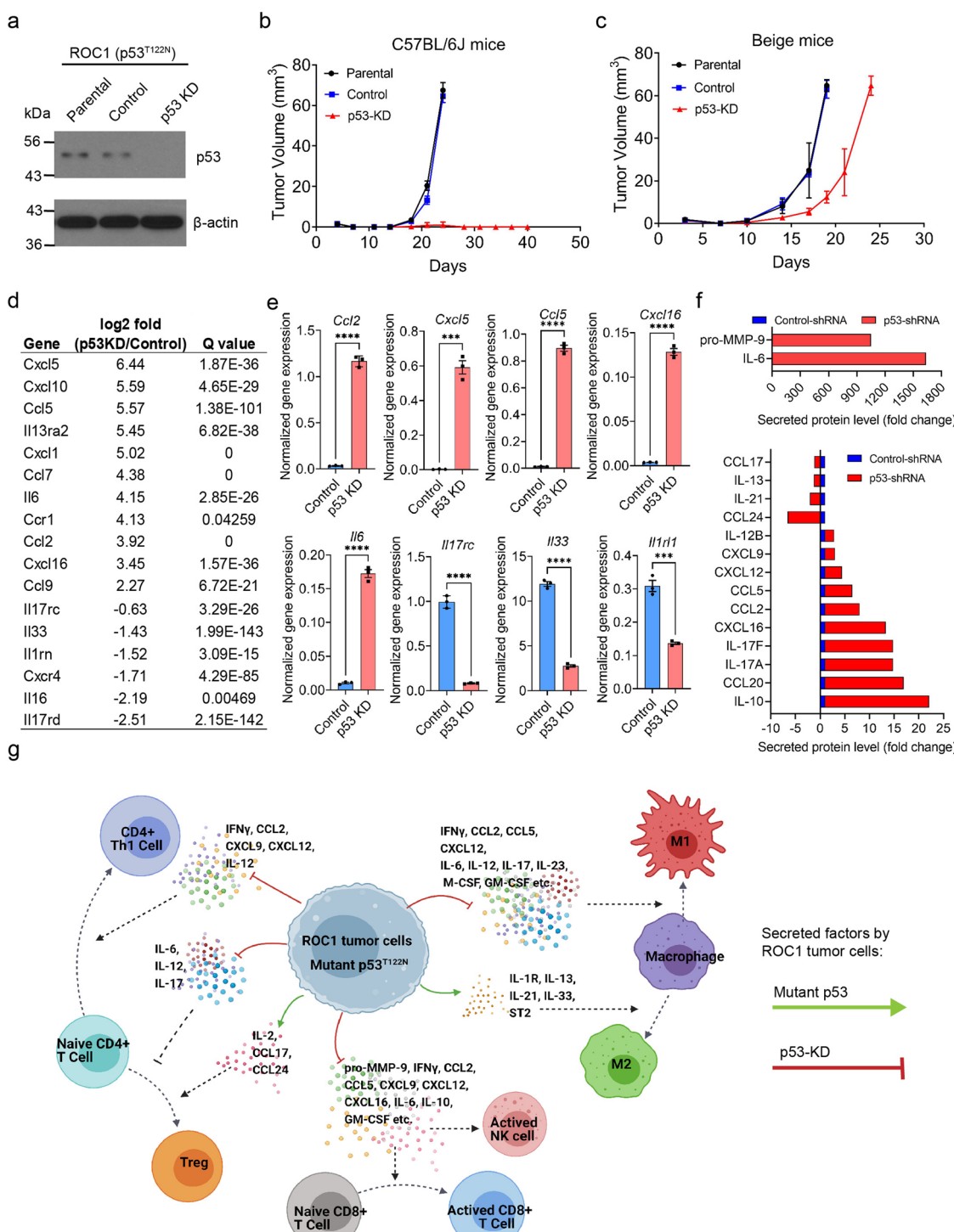

**Fig. 4 Mutant p53 in ROC1 tumors modulates tumor cell-intrinsic factors required for immune escape in immunocompetent mice. a** Western blot analysis of the parental, Control (Non-Targeting Control-shRNA), and p53-KD (p53-shRNA) clones. β-actin was used as a loading control. **b, c** ROC1 cells (50,000) were implanted into the tongues of C57BL/6J mice (n = 5) and immunodeficient beige mice (n = 5) respectively, KD of mutant p53 impairs tumor growth in immune competent mice (**b**) compared with immune deficient Beige mice (**c**) (error bars = standard deviation). **d** Differential chemokine and cytokine gene expression analysis of ROC1 p53-KD cells relative to control cells, showing genes with significant Q values. **e** Validation of cytokine and chemokine RNA expression by qPCR (Bar = mean of triplicate experiment, error bars = standard error mean). Unpaired two-tailed Student's t test p-values shown, ***P < 0.0005, ****P < 0.0001, for comparison of control vs p53-KD ROC1-tumor cells. **f** Fold changes in mouse cytokine protein expression in p53-KD tumor cells relative to control tumor cells. Antibody array assay was used to measure cytokine levels from an equal amount of supernatant of 48-h culture (details in Material and methods section). The bars represent the average of duplicate independent experiment for comparison of control vs p53-KD ROC1 cell supernatants. **g** Proposed molecular model by which mutant p53 regulates the TIME. ROC1 cells acquire a carcinogen-induced mutation in p53, yielding cytokines that promote M2 macrophage polarization and the infiltration of Tregs, thereby generating an immunosuppressive TIME by excluding effector immune cells. Created with BioRender.com.

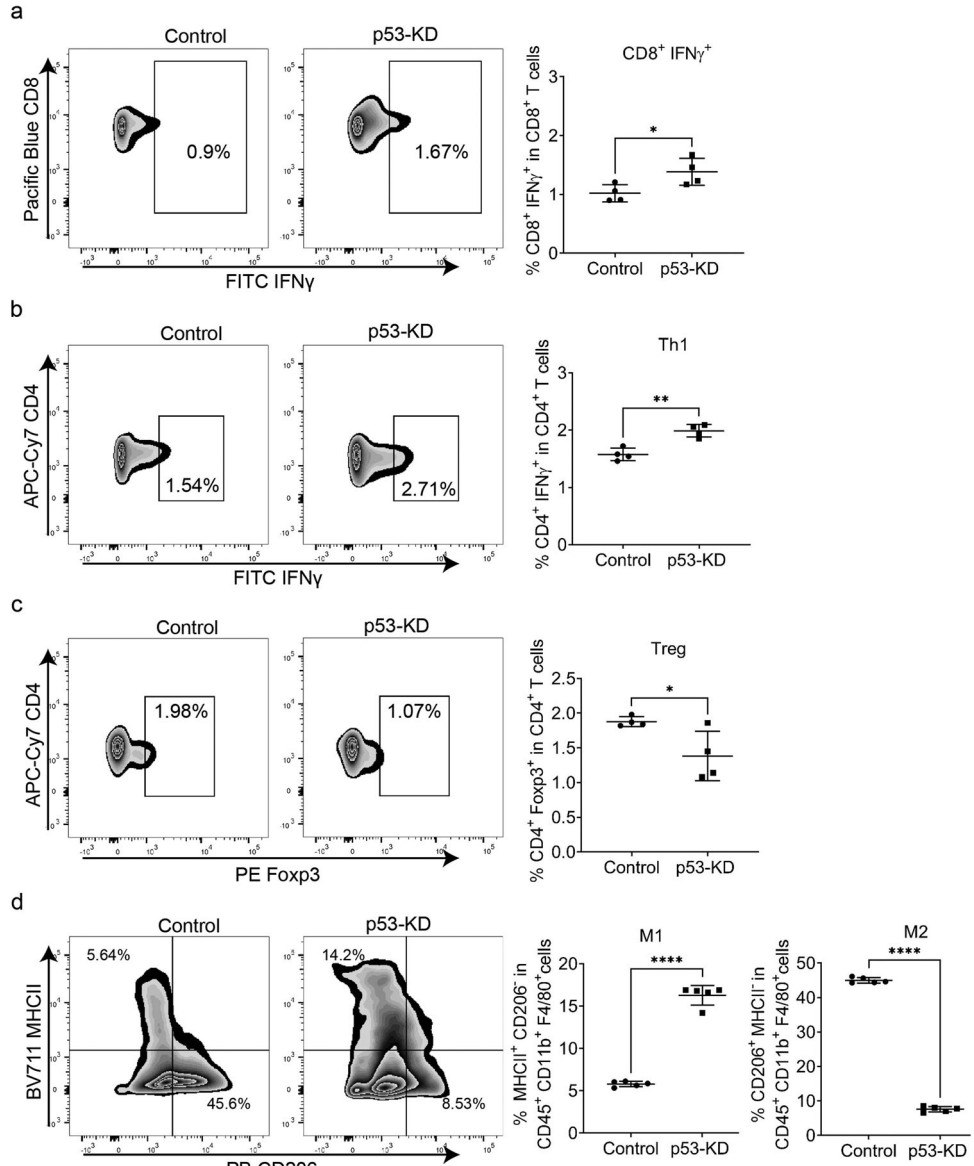

**Fig. 5 Flow cytometry of immune populations regulated by ROC1 control tumor- and ROC1 p53-KD tumor-conditioned medium.** Comparison of the frequencies of IFNγ+CD8+ T cells (**a**); IFNγ+CD4+ T cells (Th1 cells; **b**); and FOXP3+CD4+ (Tregs; **c**) in splenocytes cultured in the presence of ROC1 control tumor- or ROC1 p53-KD tumor-conditioned media. **d** Comparison of the frequencies of MHCII+CD206− (M1-type) and MHCII-CD206+(M2-type) macrophages in bone marrow precursor cells cultured in the presence of control tumor- or p53-KD tumor-conditioned medium. Dots represent individual sample ($n = 4$–5, error bars = standard deviation). Unpaired two-tailed Student's $t$ test $p$-values shown, *$p < 0.05$, **$p < 0.001$, ****$p < 0.0001$.

can be implanted orthotopically into C57BL/6J mice; have different degrees of lymph node metastasis; have different TIME characteristics; and have defined p53 mutational statuses. This set of syngeneic oral cancer cell lines provides an opportunity to study the role of mutant p53 in modulating the TIME and will be useful as a preclinical platform from which to explore alternative immunotherapies in patients with OSCC.

In this study, we developed oral cancer cell lines that had a *p53* germline GOF mutation (R172H), a *p53* LOF mutation (homozygous deletion), or *p53* mutations induced by 4-NQO, a carcinogen that mimics the tobacco-associated cancer signature[24]. The ROC3 cell line, which had a *p53* R172H germline mutation, had a higher frequency of frameshift deletions than the other three ROC cell lines did. This suggests that genomic alterations might have occurred before 4-NQO treatment was started and were enhanced with administration of the carcinogen. Moreover, C > T and C > A substitutions, a somatic mutation signature of 4-NQO[52], were

detected in all three cell lines, but ROC3 cells showed a five-fold increase in C > A and G > T substitutions, suggesting that mutant p53 R172H impairs DNA repair mechanisms, as others have reported[53].

The high frequencies of somatic single-nucleotide substitutions and frameshift deletions in the ROC3 cells strongly suggest the generation of neoantigens that might elicit a profound immune response if immune checkpoints can be overcome with treatment using checkpoint inhibitory antibodies. Similarly, the ROC1 cell line, which acquired a p53 mutation by carcinogen exposure, could form tongue tumors with fewer cells implanted and with a higher incidence of metastasis than the ROC2 and ROC3 cell lines, which had *p53* germline mutations. In this respect, other available syngeneic tumor models, such as MOC1 and MOC2 tumors, which are obtained from 7,12-dimethylbenz[a]anthracene treatment, have been extensively used; both acquire *p53* mutations and can grow orthotopically in C57BL/6 mice[54].

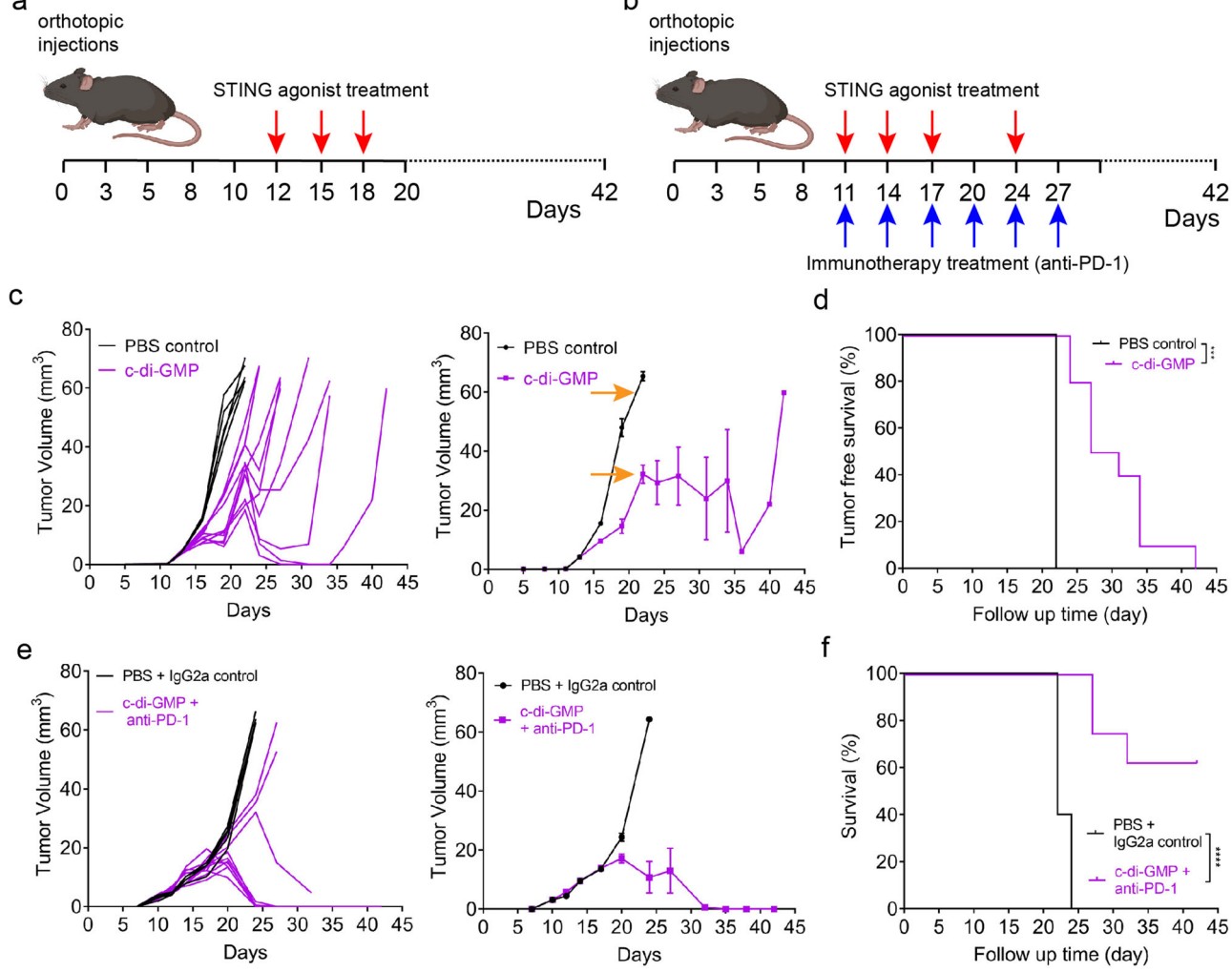

**Fig. 6 Combination immunotherapy targeting innate and adaptive immunity overcomes p53-driven tumor-mediated immune suppression. a** Process diagram of STING agonist treatment of the orthotopic ROC1-tumor model. **b** Process diagram of combined STING agonist and anti–PD-1 antibody treatment of ROC1 tumors. **c**, **d** Immune-excluded ROC1 tumors respond poorly to STING agonist monotherapy (**c**) and have reduced tumor-free survival (**d**). ***$P < 0.0002$, by Mantel–Cox test. e, f ROC1 tumors treated with STING and anti-PD-1 antibody have significantly improved tumor response (**e**) and survival (**f**). ****$P < 0.0001$, by Mantel–Cox test. Mice were injected orthotopically with 50,000 ROC1 cells control group $n = 5$, therapy group $n = 10$. The orange arrows indicate the time point selected for multiplex IHC in Fig. 7.

Interestingly, the ROC1 cell lines contain a *p53* alteration in the same genomic regions as MOC22[28]. Recently, 4MOSC cell lines were reported to share very similar characteristics with the ROC cell lines, including 4-NQO mutation signatures and mutations in cancer gene drivers (i.e., *p53*, *Fat1*, *Notch1*, *Kmt2d*, *Fat3*, and *Fat4*) in HNSCC[29].

Another study of the mutational spectrum of the 4-NQO mouse model found that mutant *Fat1* was associated with tumor grade and a high proliferation rate, whereas *Trp53* correlated with tumor grade and *Notch1* with immune infiltrate[52]. Moreover, the ROC1 tumors showed infiltration of Tregs and M2 macrophages (CD206+) and exclusion of immune effector cells, which are typical characteristics of cold tumors. In contrast, ROC2 and ROC3 warm tumors showed high infiltration of CD8+, CD4+ T cells but also high infiltration of Tregs and high expression levels of the PD-1 and TIGIT proteins, which can neutralize the immune surveillance mechanism in immunocompetent mice. We speculate that the genomic alterations in the ROC3 tumor cells might lead to the productive expression of neoantigens promoting the infiltration of CD11c+ dendritic cells, but these cells can be inactivated by the expression of immune checkpoints and Tregs. Similar results with

the 4MOSC1 cell line have been observed with use of an anti–CTLA-4 antibody[29], and both models are suitable for investigating the mechanism of immunological memory in OSCC after immunotherapy. However, because only a small fraction of patients respond to these therapies, models of immunosuppressive oral cancer need to be explored in greater mechanistic detail. We are currently studying the amount of neoantigens generated by frameshift mutations in the ROC3 cell line, which are two times greater than the amounts generated in other ROC cell lines; such immunogenomic studies might provide insights about immunoediting mechanisms in OSCC.

The TIME comprises the extracellular matrix, a combination of soluble factors, and a range of innate and acquired immune cells that impact immune surveillance and contribute to tumorigenesis, progression, and metastasis[55]. As expected, ROC1 cold tumors did not respond to ICIs, since these tumors contained infiltrated M2 macrophages and Tregs, both of which are suppressive immune cells that can exclude the infiltration and activation of effector immune cells. Similarly, 4MOSC2 tumors are strongly infiltrated with MDSC, making the tumors more resistant to ICIs[29].

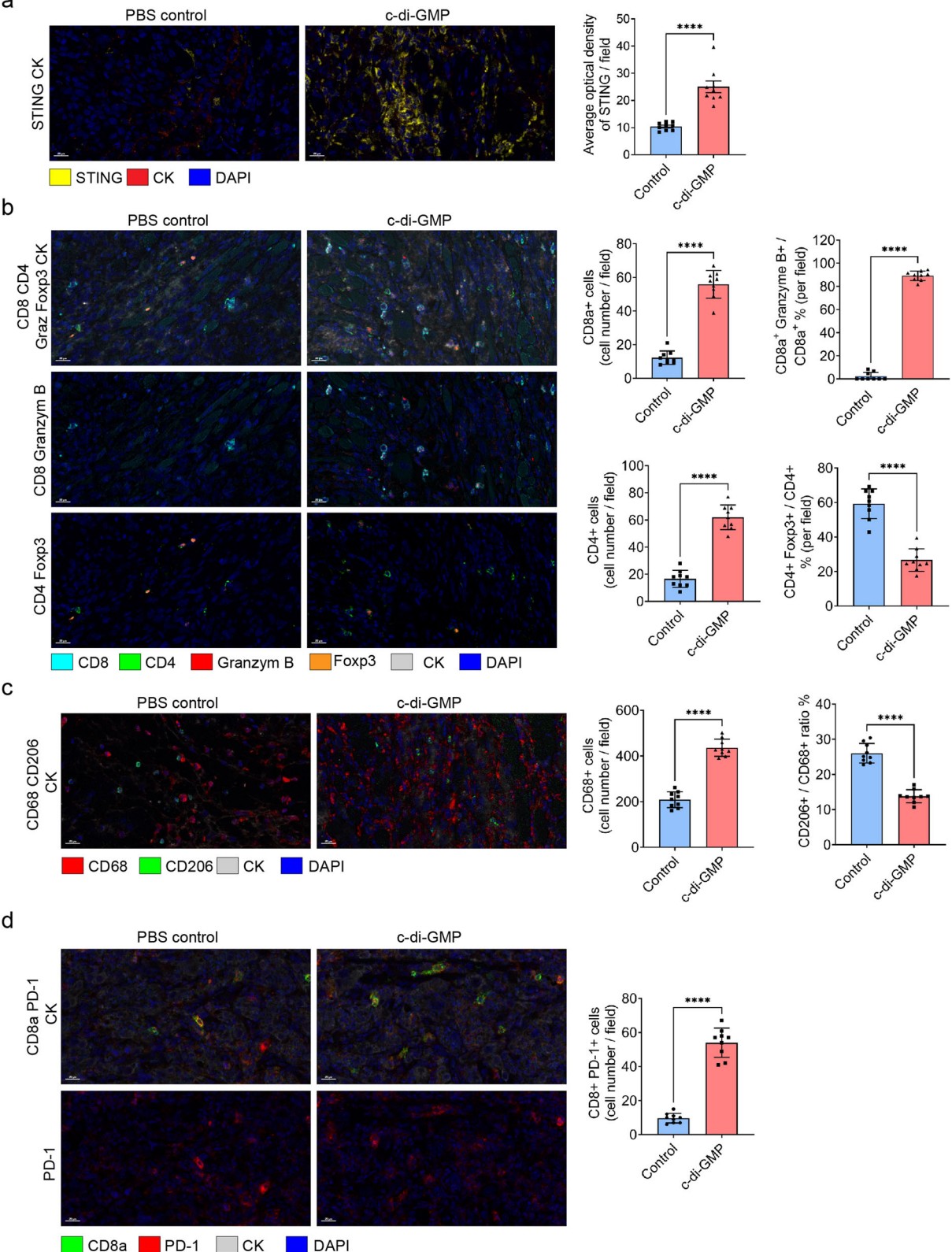

**Fig. 7 STING agonist c-di-GMP influences the TIME of ROC1 cold tumors.** Fluorescent multiplex IHC quantification analysis shows that STING (**a**), T-lymphocyte markers (CD8a, CD4, granzyme B, and Foxp3; (**b**)), macrophage markers (CD68 and CD206; (**c**)), and PD-1 (**d**) are differentially expressed between STING agonist (c-di-GMP) and PBS control groups. Representative images were processed using phenochart 1.0.12 software (scale bars, 20 μm). Multiplex IHC quantification was defined as the average optical density per view and density of cells per view (×20 magnification) and cellular density per field was quantified using Image J software ($n = 3$ slides and three random fields from each slide were quantified). Unpaired two-tailed Student's $t$ test showed the $p$-value = ****$p < 0.0001$.

Interestingly, we found that mutant p53 expression in ROC1-tumor cells is necessary to inactivate the immune surveillance mechanism to enable tumor progression. Our transcriptome and proteomic studies revealed that cytokines and chemokines are regulated by mutant p53 T122N, the expression of which is associated with the upregulation of NF-kB inhibitors (i.e., Nfkbia, Nfkbiz, and Nfkbie), which alter the canonical NF-kB pathway. These transcription factor inhibitors modulate the expression of IL-6, an inflammatory cytokine that regulates immunosuppression and immune cell polarization[56]. Recently, the IL33-ST2 pathway was shown to be involved in the immunosuppressive mechanisms resulting in cold tumors by preventing the infiltration of T cells and promoting the polarization of tumor-associated M2 macrophages, which secrete TGF-β and sustain the activation of the ST2-IL33 pathway[41]. Interestingly, high expression levels of IL33 and ST2 have been correlated with poor survival in oral carcinoma[40].

We also demonstrated that KD of mutant p53 expression in ROC1 cells disrupts the expression of IL33 and ST2 receptor, suggesting that mutant p53 mediates the transcriptional activation of this immunosuppressive mechanism. It has been shown that p53 and NF-κB have much crosstalk in the regulation of apoptosis, senescence, autophagy, epithelial mesenchymal transition, proinflammatory gene responses, and others mechanisms[57]. In addition, it has been demonstrated that mutant p53 is associated with increased NF-κB activity. Furthermore, NF-κB activation regulates cytokine pathways that sustain a chronic inflammatory state, which in turn has been associated with DNA damage, genome instability, and immunosuppressive mechanisms[17]. Thus, we hypothesize that mutant p53 can rewire NF-kB signaling pathways to sustain the IL33/ST2 signaling axis, leading to myeloid progenitor recruitment and M2 macrophage polarization. M2 macrophages secrete TGF-β1 to reinforce the immunosuppressive TIME. In addition, p53-KD in ROC1-tumor cells lead to high protein secretion levels of pro-MMP9, a metalloprotease that can digest extracellular matrix, and the chemokines IL10, CCL2, CCLl5, and CXCL16, which can attract CD8[+] T and NK cells, suggesting a strong immune response toward the ROC1 p53-KD cells in the immunocompetent mouse model[42-47]. Moreover, conditioned media from the ROC1 control and p53-KD tumor cells contained functional factors that can induce M2-to-M1 macrophage polarization, reduce CD4[+] FOXP3[+] cells (Tregs), and increase CD4[+] INFγ cells (Th1 cells). These results strongly suggest that mutant p53 drives immunosuppressive mechanisms in ROC1-tumor cells.

Cold OSCCs, which are characterized by low immune cell infiltration, are the most challenging to eradicate and are invariably associated with poor prognosis. One potential approach to overcoming the lack of a preexisting immune response, is to combine immunotherapy with a priming therapy that enhances intratumoral T-cell infiltration and response. In this study, we used the STING agonist c-di-GMP. C-di-GMP, which is a potent stimulator of innate immunity in eukaryotic organisms responsible for sensing pathogen-derived nucleic acids in the cytoplasm and subsequently activating a signaling cascade to stimulate type I and II IFN responses[51]. Our results prove that STING is positively correlated with enhanced antitumor CD8[+] and CD4[+] T-cell infiltration and migration into the tumor and inhibition of myeloid cell polarization into Tregs and M2 macrophages. The STING agonist, to some extent, overcomes immune inhibition caused by the T122N mutation and restores the immune-responsive TIME. We hypothesize that altering the pattern of cytokine and chemokine expression within the TME is a key elicitor of T-cell infiltration and response to PD-1 inhibition. Supporting our hypothesis, intratumoral delivery of the STING agonist in combination with an anti–PD-1 antibody elicited a dramatic antitumor response, a finding that is in agreement with previous studies suggesting that

STING agonists contribute to the trafficking of CD8[+] cytotoxic T cells, dendritic cells, and NK cells to the tumor site by upregulating the secretion of cytokines (e.g., CXCL9, CCL2, CCL5, CCL20, IL6)[58-61].

These results explain why the STING agonist could prolong the survival of mice with cold tumors and why the combination of the STING agonist with the anti–PD-1 antibody elicited a much better response. Previous studies showed that a STING agonist can increase PD-1 expression and cure tumors resistant to PD-1 blockade[62,63]. Our findings demonstrate that the STING agonist increases the sensitivity of ROC tumors to anti–PD-1 ICI. Similar results have been obtained with TRAMP-C2 mice, a model of aggressive prostate cancer, in which intratumoral STING delivery in combination with ICIs elicited a substantial tumor response[64].

Overall, our findings suggest that the ROC syngeneic oral cancer mouse models recapitulate the mutational landscape of human OSCC and represent both immune-excluded ("cold") tumors and immune-infiltrated tumors. These are useful models with which to elucidate the role of mutant p53 in the dysregulation of the TIME and to study immunotherapy responses in the context of clinically relevant genomic alterations to help accelerate translational research.

## Methods

**Genetically engineered mouse models.** We generated genetically engineered mouse models in which the endogenous p53 GOF mutation p53[R172H] or the p53 loss-of-function (LOF) mutation (homozygous p53 deletion) was activated or deleted in oral epithelial cells in K14-Cre mice that drive activation of the p53[R172H] and floxed-p53 conditional alleles to stratified epithelia, respectively. The following three groups of mice were generated: 1) mice with activation of p53[R172H] and deletion of the remaining p53 allele (K14 Cre[Tg/+]; p53[R172H/flox], GOF); 2) mice with homozygous deletion of p53 (K14 Cre[Tg/+]; p53[flox/flox], LOF); and 3) mice with p53 WT (K14 Cre[Tg/+]; p53[wt/wt], WT). The mutant alleles were verified by PCR with use of genomic DNA from the mice's tails as described previously[65]. To induce OSCC, we exposed 15 mice to the carcinogen 4-NQO (100 µg/mL) in drinking water with 1% sucrose (Fisher Scientific, Pittsburgh, PA) for 8 wk and monitored them weekly for tumor development. At 22-24 wk, tongue tumors were detected and harvested. OSCC-bearing mice were humanely euthanized using carbon dioxide gas chamber when they showed body weight loss of more than 20%. All mice used in this study were approved by the institutional animal care and use committee at the University of Texas – MD Anderson Cancer Center, Houston Texas, USA. (The animal experiments are described in detail in the Supplementary Materials and Methods section.) A stock solution of 4-NQO (50 mg/mL) was prepared by dissolving 4-NQO powder (Sigma-Aldrich, St Louis, MO) in DMSO and was stored at $-20\,°C$ until use.

**Establishment of syngeneic mouse oral cancer cell lines.** Mouse OSCCs on the tongue surface were harvested; half of the tissue was processed for histopathological analysis and the other half for primary tissue culture. The tumor tissue was collected aseptically and rinsed in PBS containing 500 U/mL penicillin-streptomycin. Each specimen was washed with PBS three times, transferred to culture plates, cut it into small pieces, submerged in medium (Dulbecco's modified Eagle's medium [DMEM] supplemented with 10% FBS, L-glutamine, sodium pyruvate, non-essential amino acids, 50 U/mL penicillin-streptomycin, and a vitamin solution), and incubated at 37 °C in 5% $CO_2$. Because the oral cavity is a bacterial environment, primary cell culture is prone to contamination. After 1–2 wk of primary culture, tumor cells outgrew from a tissue fragment and formed several colonies; eventually, four designated cell lines (ROC1–3) were developed. The cells were gradually expanded to larger plates, and epithelial cells were enriched by differential trypsinization for fibroblast removal.

ROC1 cells were from the K14Cre; p53[wt/wt] mouse; ROC2 cells were from the K14Cre; p53[f/f] mouse; and ROC3 cells were from the K14Cre; p53R172H[CA/f] mouse. Next, 5 million ROC1–3 cells were separately implanted subcutaneously into the C57BL/6J mice; only ROC1 successfully generated subcutaneous tumors. Non-tumorigenic cells in the C57BL/6J mice (ROC2 and ROC3) were injected subcutaneously (5 million cells each) into athymic nude mice, which generated skin tumors. These skin tumors were processed for tissue culture, and differential trypsinization was performed for fibroblast removal from primary cultures. At passages 5–7, ROC1, 2, and 3 cells were labeled with phycoerythrin-conjugated anti-epidermal growth factor receptor (EGFR) antibody (GeneTex clone ICR10) and sorted by flow cytometry. Next, the enriched EGFR[+] cells were expanded and subcultured for soft agar colony formation assay (Supplementary Fig. 1a). To ensure that the final cell line was originally from a single tumor cell colony, the biggest tumor colony in soft agar was picked out under microscopy for further tissue culture. (The Soft agar colony formation assay is described in detail in the

Supplementary Materials and "Methods" section). The ROC cell lines are from mouse origin confirmed by CellCheck 19-mouse STR profile by IDEXX BioAnalytics (Supplementary Table 5). Cell lines are available upon request.

**In vitro assays**. The colony formation, cell proliferation, wound scratch, and Matrigel invasion assays are described in the Supplementary Materials and Methods section.

**In vivo subcutaneous and orthotopic mouse model and treatment**. Female beige mice (Jackson Laboratory, #000629) and C57BL/6J mice (Jackson Laboratory, #000664), age 8–10 wk, were housed in a specific pathogen–free animal facility. All animal experimentation was approved by MD Anderson's Institutional Animal Care and Use Committee. The animal experiments are described in detail in the Supplementary Materials and Methods section.

**Immunohistochemical analysis**. Sections were prepared from formalin-fixed paraffin-embedded (FFPE) specimens of mouse tumor tissues and subjected to immunohistochemical (IHC) staining with the indicated antibodies. IHC quantification was defined as the density of cells and average optical density per view (20X; $n = 3$ stained tumors; every antibody staining had 5 field views). The antibodies used for IHC are described in the Supplementary Materials and Methods section.

**Fluorescent multiplex IHC consecutive staining on a single slide**. Slides were prepared from FFPE specimens of mouse tumor tissues. Multiplex IHC consecutive staining on a single slide was performed with use of the Opal 7-Color Manual IHC Kit (AKOYA Biosciences, #NEL811001KT) to analyze the TIME. A series of sequential cycles of staining, image scanning, and destaining of chromogenic substrate was performed on FFPE tissue samples. Visualization of 7-Color Opal slides was performed by using a Mantra or Vectra quantitative pathology imaging system. Each system uses multispectral imaging for quantitative unmixing of many fluorophores and tissue autofluorescence. The antibodies used for multiplex IHC are described in the Supplementary Materials and Methods section.

**DNA and RNA isolation**. ROC1-3 cells genomic DNA was isolated with use of a Blood and Cell Culture DNA Mini Kit (Qiagen, #13323) according to the manufacturer's protocol. Total cells (ROC1–3 cells, ROC1 p53-shRNA cells, and ROC1 control-shRNA cells) and tumor tissue (ROC1-3 tongue tumors) RNA was isolated with use of RNA Miniprep Plus (Zymo Research, Irvine, CA) according to the manufacturer's instructions. Genomic DNA and RNA concentrations were determined by using the NanoDrop system (Thermo Scientific, Wilmington, DE). Whole-exome sequencing and RNA-seq data were generated by the BGI Genomics Company. The RNA-seq reads were mapped against the reference genome of GRCm38 of Mus musculus strain C57BL/6J by using TopHat2 (v2.1.1), and the mapped reads per gene were counted by using HTSeq (v0.11.0) based on gene annotation of GENCODE M19 (Supplementary Table 6). The reads counts were scaled/normalized by transcripts per million (TPM). WES reads were mapped by using bwa (with default parameter setting) to against the mouse genome assembly mm10. Nucleotide mutations were called by using Varscan2 (parameter setting:-strand-filter 1-min-coverage 30-p-value 0.01-min-freq-for-hom 0.9–output-vcf 1-variants 1) based on the output generated by samptools mpileup of bam files. The VCF files of mutations for down-streaming analysis were annotated by the refGene database and filtered by the snp142 database.

**Generation of stable shRNA KD cells**. We obtained p53-shRNA and non-targeting control (NTC) shRNA lentiviral glycerol stock clones from MD Anderson's Functional Genomics Core Facility. The p53 shRNA clone (v3lhs_646511) had the corresponding nucleotide sense sequence CACTACAAGTACATGTGTA. Glycerol stocks were plated in ampicillin LB agar and incubated at 37 °C overnight. Single bacteria colonies were grown in LB media for plasmid DNA purification by using a Midiprep kit (Qiagen). Briefly, lentiviral plasmids were mixed with Lipofectamine 2000 (ThermoFisher Scientific, Carlsbad, CA) and transfected into 293FT cells for 8 h in serum-free media. At 48 h after transfection, media containing the virus was collected and centrifuged at 1200 rpm to remove cellular debris. ROC1 cells were grown at 70% confluence and infected with the virus-containing media supplemented with polybrene (1 μg/mL) for 24 h. After 48 h, cells were plated at 50% confluence and selected with puromycin (2 μg/mL) in complete DMEM for 7 d or until all non-infected control cells were dead. Next, ROC1 cells stably expressing p53-shRNA or NTC shRNA were enriched for green fluorescent protein expression by using flow cytometry under sterile conditions to improve the p53-KD.

**Western blotting**. Whole-cell lysates were prepared, and Western blotting was conducted as described previously[66]. Primary antibodies against p53 (Cell Signaling, 1:1000, #32532) and β-actin (Santa Cruz Biotechnology, 1:5000, #sc81178) were used.

**Reverse transcription-quantitative PCR (RT-qPCR)**. Total RNA was purified and DNase-treated by using the RNeasy Mini Kit (Qiagen). Synthesis of cDNA was performed by using SuperScript IIII reverse transcriptase according to the manufacturer's protocol (Life Technologies). We used Bio-Rad CFX96 Touch Real-Time PCR Detection System and SsoAdvanced Universal SYBR Green Supermix (Bio Rad, USA). The RT-qPCR conditions were as follows: Initial denaturation at 94 °C for 2 min; 40 cycles of denaturation at 94 °C for 15 s, and annealing and extension at 60 °C for 1 min. The relative expression level of RNA was calculated with use of the following formula: $RQ = 2^{-\Delta\Delta Cq}$. Each reaction was performed in triplicate. All signals were normalized by using Gapdh gene as an internal expression control. The primers used are shown in Supplementary Table 7.

**Cytokine profile of conditioned medium**. To prepare serum-free or low-serum medium samples, we seeded 1 million NTC shRNA and p53-shRNA ROC1 cells into 100-mm tissue culture plates with complete medium. After 72 h, we replaced the medium with 6 mL of low-serum medium (0.2% FBS). After another 48 h, the supernatants were collected and centrifuged at 2,000 rpm at 4 °C for 10 min. Finally, 0.2 μ filtered ROC1 cell supernatants were frozen and stored at −80 °C. Cytokine profiling was performed by RayBiotech by using the Quantibody Mouse Cytokine Array Q1000 (Peachtree Corners, GA).

**Collection of conditioned cell medium**. ROC1 NTC shRNA and p53-shRNA cells were plated at a density of 4000 cells/cm² in complete DMEM. After 48 h, the culture medium was refreshed with complete medium. Approximately 48 h later, the cell medium was harvested and centrifuged at 4000g for 10 min. Supernatants were stored at −80 °C.

**Isolation and flow cytometry of murine bone marrow cells and splenocytes**. BMCs and splenocytes were isolated from 8- to 12-wk-old C57BL/6J mice. The BMCs were harvested from femurs and tibias flushed with a 25 G needle and filtered through a 70-μm cell strainer. Splenocytes were isolated by straining a mashed spleen through a 70-μm cell strainer. Red blood cells were removed by using red blood cell lysis buffer (eBioscience, #00-4333-57) according to the manufacturer's recommendations. BMCs and splenocytes were seeded separately at densities of $(1–2) \times 10^5$ cells/cm² and $(1–2) \times 10^6$ cells/cm², respectively, and cultured with the cell-conditioned medium. After 36 h, the conditioned medium was refreshed with additional conditioned media. Finally, after another 36 h, BMCs and splenocytes were harvested and suspended in a fluorescence-activated cell sorting buffer for cell surface staining. Antibody staining was conducted by incubating the cells with antibodies for 30 min on ice in the presence of mouse 2.4G2 monoclonal antibody (Tonbo, #70-0161-U100) to block FcγR binding. For Foxp3 and granzyme B staining, a transcription factor–staining kit (Invitrogen, #00-5523-00) was used. To assess cytokine production, T cells were stimulated with 50 ng/mL phorbol 12-myristate 13-acetate (Sigma, #P148), 500 ng/mL ionomycin (Sigma, #407950), 1 μL/mL Golgi-Stop (BD Biosciences, #554724), and 1 μL/mL Golgi-Plug (BD Biosciences, #555029) for 4 h at 37 °C. T cells were subsequently stained for cell surface markers before intracellular cytokine staining. The antibodies used are shown in Supplementary Table 8. All data were acquired with use of an LSRFortessa X-20 flow cytometer (BD Biosciences) and analyzed with FlowJo software (Tree Star).

**Statistics and reproducibility**. The numerical data were represented as the mean ± SEM. Statistical p-values between two groups were calculated using the unpaired two-tailed Student's t test. Comparisons of more than two groups were assessed by one-way ANOVA with Tukey's post hoc test. Tumor-free survival was assessed by the Mantel–Cox test. Statistical significance was assigned at a p-value less than 0.05. All results were independently reproduced at least three times with similar results.

**Reporting summary**. Further information on research design is available in the Nature Research Reporting Summary linked to this article.

## Data availability

The source of data for the graphs and charts in the main figures as well as the original uncropped blot/gel images are provided in Supplementary information. All other data generated during the current study are available from the corresponding author on reasonable request. Publicly available RNAseq datasets of the ROC1 p53 knockdown are at Gene Expression Omnibus (GEO), study accession number GSE201722.

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

## Acknowledgements

The authors thank Dr. Jared Burks for help in the multiplex IHC imagining and cell sorting at the Flow Cytometry and Cell Imaging Core Facility, Dr. Fernando Benavides at the Laboratory Animal Genetic Services, the Functional Genomics and Advance Technology Genomics Core Facilities at MD Anderson Cancer Center, for supportive reagent and service supported by the NIH/NCI under award number P30CA016672. Illustrations in the figures were created with BioRender. This work was supported by The Mrs. Nancy L. De Anda Research Foundation and Mary K. Chapman Foundation (to J.N.M.).

## Author contributions

Y.S., A.G.S., J.N.M., and R.R. made substantial contributions to the conception, design, acquisition, analysis, and interpretation of the data, drafted and substantively revised the work. B.W., R.W., and Y.C. have generated and interpretation of the data, and substantively revised the work. F.O.G.N., X.T., J.W., and C.R.P. have performed analysis and interpretation of sequencing data and substantively revised the work. T.X., B.Y., A.E.R.R., X.R., and A.A.O. have contributed significantly to the acquisition, analysis, and interpretation of the data, and substantively revised the work. All authors have reviewed and approved the submitted version of the manuscript and are accountable for their own contributions to the manuscript.

## Competing interests

The authors declare no competing interests.
