## [Peer Review File · Communications Biology]

Reviewers' comments:

Reviewer #1 (Remarks to the Author):

1) Please add a reference for ROC cell line since i found this on pubmed (10.1007/978-1-4684-5823-7_35 - Roc-1) which may bring in future to miss-understanding.

2) This statement "We used the ROC1 cell line, which has a relatively "cold" TIME, to investigate the effect of mutant p53 on the modulation of cell-intrinsic factors that shape the immune landscape and impact sensitivity to immunotherapy" is confusing in the introduction.

3) The aim of the study and study model is unclear at the end of the introduction. Add few short sentences

4) In the introduction, please add few statement about the importance of TIME and survival of patients (<https://doi.org/10.1002/cam4.3440>)

(<https://doi.org/10.1016/j.oraloncology.2021.105420>)

5) Please, include this reference here Bioinformatic studies detected acquired mutations in p53, Notch1,

Csmd3, Med12l, Kmt2d, Fat1, Fat3, Fat4, Akap9, and Myh6 in murine ROC cell lines, as reported previously in human HNSCC (Fig. 1D), and discuss its results from supplemental material about mutational status and immune-related pathways

6) report p-values in the manuscript for statistically significant results and name of statistical test you performed between ().

Reviewer #2 (Remarks to the Author):

The study investigated the effect of mutant p53 on the modulation of cell-intrinsic factors that shape the immune landscape and impact sensitivity to immunotherapy. The authors generated four syngeneic mouse oral cancer cell lines (ROCs) with p53 mutations to better understand the role of epithelial cells expression mutant p53 in shaping the TIME.

The study is of interest to the journal's readers and has presented establishment of new murine oral cancer cell lines.

1-Head and neck squamous cell carcinoma is a heterogeneous group of tumours that are associated with a variable list of risk factors and is not considered the sixth most common cancer. The study has focused on a subgroup which is oral cavity SCC. The mouse model of oral carcinogenesis using 4-NQO mimics the effects induced by tobacco exposure. Thus, oral cavity SCC should replace HNSCC in the title and throughout the text of the manuscript.

2- Concerning the validation of the murine oral carcinogenesis model with human oral SCC, the authors are advised to benefit from the previous work of Sequeira I, Rashid M, Tomás IM, et al. Genomic landscape and clonal architecture of mouse oral squamous cell carcinomas dictate tumour ecology. *Nat Commun.* 2020;11(1):5671. Published 2020 Nov 9. doi:10.1038/s41467-020-19401-9.

Introduction

2- Further, according to GLOBOCAN 2020 data, lip and oral cavity cancers are ranked 18th with an incidence of 377,713 and mortality of 177,757. The epidemiology statements should be revised to reflect these data and the authors should cite a recent paper on global cancer epidemiology (Sung H, Ferlay J, Siegel RL, et al. *Global Cancer Statistics 2020: GLOBOCAN Estimates of Incidence and Mortality Worldwide for 36 Cancers in 185 Countries.* *CA Cancer J Clin.* 2021;71(3):209-249. doi:10.3322/caac.21660).

3- Please cite the statement "Immune checkpoint inhibitors (ICIs),, including OSCC." using the following reference: Kujan O, van Schaijik B, Farah CS. *Immune Checkpoint Inhibitors in Oral Cavity Squamous Cell Carcinoma and Oral Potentially Malignant Disorders: A Systematic Review.* *Cancers (Basel).* 2020;12(7):1937. Published 2020 Jul 17. doi:10.3390/cancers12071937

4- Please use the following reference (Olivier M, Hollstein M, Hainaut P. TP53 mutations in human

- cancers: origins, consequences, and clinical use. Cold Spring Harb Perspect Biol. 2010;2(1):a001008. doi:10.1101/cshperspect.a001008) to cite the statement "Somatic TP53 mutations, the most common alterations among all cancers,"
- 5- Please cite the following statement "Overall, genomic alterations in TP53 can not only contribute to tumorigenesis by driving the growth and survival of the epithelial tumor compartment but also impact the TIME to enable evasion of the immune response mechanism in different cancer types". Results:
 - 6- The statement "carcinogenesis is considered the most representative OSCC model" has a typo and 4-NQO oral carcinogenesis is not the most representative as HPV-related carcinogenesis is not considered. Please correct to read "4-NQO-induced chemical carcinogenesis is representative of OSCC model.
 - 7- Please provide details on the administration of 4-NQO to mice and correct the typo in the units. It should read "100mg/mL".
 - 8- Please delete the first paragraph from the discussion as it is already covered in the introduction. Discussion
 - 9- The inflammatory crosstalk between NF- κ B and p53 should be expanded.
 - 10- The manuscript contains several typos and a proofread of the text is recommended.

Reviewer #3 (Remarks to the Author):

Shin et al developed novel oral squamous cell carcinoma (OSCC) mice models with defined Tp53 mutations and characterized the tumor immune microenvironment (TIME) associated with different degrees of immune checkpoint inhibitor (ICI) responsiveness. They focused on ROC1 cell line harboring a T122N amino acid change of the Tp53 and revealed that the mutation promotes a cold TIME enriched with immunosuppressive M2 macrophages highly resistant to ICI therapy. ROC1 cold tumors responded to combination immunotherapy.

The results are clearly presented and this study seems to provide an important contribution to development effective immunotherapeutic approaches for not only OSCC, but also other ICI resistant solid tumors. However, the following questions are pertinent:

1) Is only T122N mutation of the Tp53 gene essential for the cold TIME? As far as I know, mouse T122N corresponds to human T125N mutation, and the frequency of T125N mutation was zero according to the TCGA database. The authors claimed that this mutation was the same genomic region as MOC22, but the MOC22 also had K384* mutation, likely pathogenic variant. I would like to recommend to the authors to confirm that nothing has been overlooked regarding the Tp53 mutation, such as splice site mutation, on ROC1 cell line.

2) Don't the ROC tumors have any other significantly mutated genes than genes listed in Figure 1D? According to the TCGA data, genes such as CDKN2A, PIK3CA, PCLO, NSD1, CASP8, KMT2C, and EP300, are also frequently mutated in head and neck squamous cell carcinomas, and some of them were reported as somatic variants correlated to the ICI scores (<https://pubmed.ncbi.nlm.nih.gov/33230435/>).

Other comments are followings:

Page 7, "Next, we performed whole-exome sequencing of the four ROC lines with excellent coverage mapping using the GRCm38 (mm10) C57BL/6J mouse genome as a reference (Supplementary Table 3).";

3) The authors should describe the method for whole-exome sequencing, coverage numbers, and mean coverage depth numbers.

Page 8, "Furthermore, we assessed the association between the ROC cell line mutations and the most significantly mutated genes in The Cancer Genome Atlas (TCGA)-HNSCC cohort. Bioinformatic studies detected acquired mutations in p53, Notch1, Csm3, Med12l, Kmt2d, Fat1, Fat3, Fat4, Akap9, and Myh6 in murine ROC cell lines, as reported previously in human HNSCC (Fig. 1D)." and Figure 1D;

4) The authors should add each variant allele frequency and depth.

5) Some genes are not defined as "cancer gene" by OncoKB database. Please revise the gene list.

6) Please describe how to calculate the mutation frequency of TCGA-HNSCC because the numbers are seemed to be incorrect.

7) Did ROC4 have Q61H mutation in Kras not Hras?

Page 10, "As described above, ROC1 cells acquired a p53 mutation; such an alteration is detected in 85% of HNSCC patients, suggesting that p53 mutations influence the TIME."

8) The authors described "85%" in this sentence, inconsistent to Figure 1D. Please describe the source of this frequency.

Page 11, "We next performed RNAseq of ROC1 tumors grown in immunocompetent C57BL/6J mice to assess immune infiltration and identify tumor cell-intrinsic factors that might promote tumorigenesis and immunosuppression mechanisms."

9) The authors should describe the method for RNA-seq and number of uniquely mapped reads.

Page 11, "Bioinformatics analysis revealed that ROC1 tumors expressed markers of macrophages and Tregs as well as immune checkpoints associated with these cell types (Supplementary Fig. 5A)."

10) What did the authors compare?

11) I'm interested in RNA-seq results in other ROC cells. Please mention why the authors didn't subject those cells for RNA-seq.

Page 11, "We assessed the transcriptome analysis in ROC1 p53-KD over control cells and revealed a dramatic enrichment of proinflammatory cytokines and chemokines confirmed by gene set enrichment analysis (Fig. 4D)."

12) Fig. 4D seems like indicating the result of differentially expressing genes, not of gene set enrichment analysis.

13) Please describe the method for this analysis.

Page 12, "In ROC1 p53-KD cells, IL33 and ST2 receptor were both downregulated (Fig. 4E); the IL33/ST2 signaling pathway has been reported to sustain tumor-associated macrophages and activate tumor growth factor-beta (TGF- expression to sustain a signaling loop to promote cancer progression."

14) "TGF- " is garbled.

Page 13, "We exposed BMCs and splenocytes to ROC1 control- and ROC p53-KD-conditioned media for 72 hours and identify the different immune cell populations through flow cytometry (Supplementary Fig. 6 and 7)."

15) "BMCs" should be replaced to "bone marrow cells (BMCs)".

Page 24, "Genomic DNA and RNA concentrations were determined using the NanoDrop system (Thermo Scientific, Wilmington, DE)."

16) In case the genomic DNA was subjected to whole-exome sequencing, dsDNA quantification using such as Qubit would be required. As for RNA-seq, the assessment of RNA integrity is a critical.

Page 25, "Reverse transcription-quantitative PCR (RT-qPCR). We used Bio-Rad CFX96 Touch Real-Time PCR Detection System and SsoAdvanced Universal SYBR Green Supermix (Bio Rad, USA) ... Each reaction was performed in triplicate."

17) The authors should describe the method for reverse transcription reaction and internal control gene name (Gapdh).

18) The number of references is out of order.

19) Page 5 of Supplementary Materials and Methods; "Western blotting" section was duplicate with "Methods".

Point-by-point-reply

Reply to Reviewer #1

We thank reviewer #1 for his/her careful review and are grateful for the opportunity to further improve our manuscript. We address his/her comments as follows:

1) *“Please add a reference for ROC cell line since i found this on pubmed (10.1007/978-1-4684-5823-7_35 - Roc-1) which may bring in future to miss-understanding.”*

We have included the information as requested. **(Line 94, p.5; Ref 20)**

2) *“This statement “We used the ROC1 cell line, which has a relatively “cold” TIME, to investigate the effect of mutant p53 on the modulation of cell-intrinsic factors that shape the immune landscape and impact sensitivity to immunotherapy” is confusing in the introduction.”*

We have edited this paragraph as suggested **(Line 99-101, p.6)**

3) *“The aim of the study and study model is unclear at the end of the introduction. Add few short sentences.”*

We have included a brief relevance description of our study, as indicated by the reviewer **(Line 101-104, p.6)**

4) *“In the introduction, please add few statement about the importance of TIME and survival of patients (<https://doi.org/10.1002/cam4.3440>) (<https://doi.org/10.1016/j.oraloncology.2021.105420>)”*

We have included the references suggested as well some sentences at the introduction. **(Line 72-76, p.4-5, Ref. 11, 12)**

5) *“Please, include this reference here Bioinformatic studies detected acquired mutations in p53, Notch1, Csmc3, Med12l, Kmt2d, Fat1, Fat3, Fat4, Akap9, and Myh6 in murine ROC cell lines, as reported previously in human HNSCC (Fig. 1D), and discuss its results from supplemental material about mutational status and immune-related pathways.”*

We now include references for these mutations in other murine oral cancer cell lines **(Line 161, p. 9, Ref. 28-33)**.

The reviewer brings up an interesting point about the mutational status and the immune related pathways. ROC3 cell line shows two-fold more frame shift deletions, suggesting increased production of neoantigens generated by the tumor cells. As shown in Figure 1E, the ROC3 cell

line injected at 100,000 cells undergoes tumor size reduction at day 10, however by day 25, the tumor size increases suggesting that the tumor undergoes immune escape. We have repeated this experiment showing reproducibility and raising the possibility that tumor cells may escape immune-mediated destruction through loss of immunogenic neoantigens. We are currently carrying out experiments to test this hypothesis, and have included a brief discussion about these ongoing studies in the discussion section (**Line 377-380, p. 18**)

6) *“report p-values in the manuscript for statistically significant results and name of statistical test you performed between ().”*

We have included the information requested in the figure legend of the manuscript, following Journal format.

Point-by-point-reply**Reply to Reviewer #2**

We thank reviewer #2 for his/her careful review and are grateful for the opportunity to further improve our manuscript. We are pleased that he/she feels that *“The study investigated the effect of mutant p53 on the modulation of cell-intrinsic factors that shape the immune landscape and impact sensitivity to immunotherapy. The authors generated four syngeneic mouse oral cancer cell lines (ROCs) with p53 mutations to better understand the role of epithelial cells expression mutant p53 in shaping the TIME. The study is of interest to the journal's readers and has presented establishment of new murine oral cancer cell lines.”*

1- *“Head and neck squamous cell carcinoma is a heterogeneous group of tumours that are associated with a variable list of risk factors and is not considered the sixth most common cancer. The study has focused on a subgroup which is oral cavity SCC. The mouse model of oral carcinogenesis using 4-NQO mimics the effects induced by tobacco exposure. Thus, oral cavity SCC should replace HNSCC in the title and throughout the text of the manuscript.”*

We agree with the reviewer and we have made all the changes in the title and manuscript text (**Line 2, p. 1**).

2- *“Concerning the validation of the murine oral carcinogenesis model with human oral SCC, the authors are advised to benefit from the previous work of Sequeira I, Rashid M, Tomás IM, et al. Genomic landscape and clonal architecture of mouse oral squamous cell carcinomas dictate tumour ecology. Nat Commun. 2020;11(1):5671. Published 2020 Nov 9. doi:10.1038/s41467-020-19401-9.”*

We agree with the reviewer that this a great resource for all of us working with this carcinogenesis mouse model which is reference in our manuscript (**Line 362-364, p. 18; Ref.50**).

Introduction

2- *“Further, according to GLOBOCAN 2020 data, lip and oral cavity cancers are ranked 18th with an incidence of 377,713 and mortality of 177,757. The epidemiology statements should be revised to reflect these data and the authors should cite a recent paper on global cancer epidemiology (Sung H, Ferlay J, Siegel RL, et al. Global Cancer Statistics 2020: GLOBOCAN Estimates of Incidence and Mortality Worldwide for 36 Cancers in 185 Countries. CA Cancer J Clin. 2021;71(3):209-249. doi:10.3322/caac.21660).”*

We included this information as well the reference indicated (**Line 58-59, p. 4; Ref. 5**)

3- *“Please cite the statement “Immune checkpoint inhibitors (ICIs),, including OSCC.” using the following reference: Kujan O, van Schaijik B, Farah CS. Immune Checkpoint Inhibitors in Oral Cavity Squamous Cell Carcinoma and Oral Potentially Malignant Disorders: A Systematic Review. Cancers (Basel). 2020;12(7):1937. Published 2020 Jul 17. doi:10.3390/cancers12071937”*

We have included the reference indicated (**Line 64-66 p. 4, Ref. 7**)

4- *“Please use the following reference (Olivier M, Hollstein M, Hainaut P. TP53 mutations in human cancers: origins, consequences, and clinical use. Cold Spring Harb Perspect Biol. 2010;2(1):a001008. doi:10.1101/cshperspect.a001008) to cite the statement “Somatic TP53 mutations, the most common alterations among all cancers,”*

We have added the reference suggested (**Line 77 p. 5, Ref. 13**)

5- *“Please cite the following statement “Overall, genomic alterations in TP53 can not only contribute to tumorigenesis by driving the growth and survival of the epithelial tumor compartment but also impact the TIME to enable evasion of the immune response mechanism in different cancer types”.*

We have included a reference for the statement indicated (**Lines 87-90, p.5, Ref 15**)

Results:

6- *The statement “carcinogenesis is considered the most representative OSSC model” has a typo and 4-NQO oral carcinogenesis is not the most representative as HPV-related carcinogenesis is not considered. Please correct to read “4-NQO–induced chemical carcinogenesis is representative of OSCC model.*

We have edited the sentence indicated (**Line 109, p. 6**)

7- *Please provide details on the administration of 4-NQO to mice and correct the typo in the units. It should read “100mg/mL”.*

We have revised the concentration used in our experiments and others (**Line 112, p. 6; Line 468-469, p. 22**)

8- *Please delete the first paragraph from the discussion as it is already covered in the introduction.*

We have deleted the first paragraph in the discussion section as suggested by the reviewer (**p.16**)

Discussion

9- The inflammatory crosstalk between NF- κ B and p53 should be expanded.

We have expanded our information in regard to the interplay of p53 and NF- κ B (**Lines 402-408, p. 19-20**)

10- The manuscript contains several typos and a proofread of the text is recommended.

The manuscript has been revised and edited accordingly.

In summary, we greatly appreciate the thoughtful feedback from the reviewers, and have made all requested changes, resulting in a greatly improved manuscript. We hope that the present version will be found acceptable for publication in Communications Biology.

Point-by-point-reply**Reply to Reviewer #3 (Remarks to the Author):**

We thank reviewer #3 for his/her careful review and are grateful for the opportunity to further improve our manuscript. We are pleased that he/she feels that “*Shin et al developed novel oral squamous cell carcinoma (OSCC) mice models with defined Tp53 mutations and characterized the tumor immune microenvironment (TIME) associated with different degrees of immune checkpoint inhibitor (ICI) responsiveness. They focused on ROC1 cell line harboring a T122N amino acid change of the Tp53 and revealed that the mutation promotes a cold TIME enriched with immunosuppressive M2 macrophages highly resistant to ICI therapy. ROC1 cold tumors responded to combination immunotherapy. The results are clearly presented and this study seems to provide an important contribution to development effective immunotherapeutic approaches for not only OSCC, but also other ICI resistant solid tumors.*”

However, the following questions are pertinent:

1) “*Is only T122N mutation of the Tp53 gene essential for the cold TIME? As far as I know, mouse T122N corresponds to human T125N mutation, and the frequency of T125N mutation was zero according to the TCGA database. The authors claimed that this mutation was the same genomic region as MOC22, but the MOC22 also had K384* mutation, likely pathogenic variant. I would like to recommend to the authors to confirm that nothing has been overlooked regarding the Tp53 mutation, such as splice site mutation, on ROC1 cell line.*”

We agree with the reviewer comment, MOC22 also contains a *Trp53* nonsense mutation at position K384*. We had performed Sanger sequencing of mouse *Trp53* and we have confirmed the T122N mutation in the ROC1 cell line. As note, there are reports of other murine oral cancer cell lines generated by using K14-EGFP-miR-211 transgenic mice exposed with 4NQO carcinogen. Interestingly, the tumorigenic cell line MOC-L1 in the study also selected a mutation in p53 T122N, which is tumorigenic and metastatic in a C57BL/6 syngeneic mouse model (PMID: 30922255). The T125 mutation is located in the sheet-loo-helix motif stabilizer region of p53 which correlated in some patient groups with poor prognosis in ovarian and breast cancer (PMID: 26215675). Other studies using UV skin mouse models have identified a novel p53 mutational hotspot (T122L) which alters transactivation functions (PMID: 12173040). While this mutation is not frequently detected in oral cancer, the role of this mutation in p53 might have an important impact in the regulation of the tumor microenvironment as described in our study.

2) “*Don't the ROC tumors have any other significantly mutated genes than genes listed in Figure 1D? According to the TCGA data, genes such as CDKN2A, PIK3CA, PCLO, NSD1, CASP8, KMT2C, and EP300, are also frequently mutated in head and neck squamous cell carcinomas, and some of them were reported as somatic variants correlated to the ICI scores*” (<https://pubmed.ncbi.nlm.nih.gov/33230435/>).

We did not identified any mutation in other frequent mutated HNSCC genes like: *Cdkn2a, Pik3a, Pclo, Nsd1 Casp8, Kmt2c, Ep300, Jub, Eph2, B2m, Flg, Nfe2l2, Fbxw7, Znf750, Hras, Necab1,*

Rb1, Tgfbr2, Ctcf, Rac1, Steap4, Prb1, Cul3, Plscr4, Krt5, Fcrl4, Slc26a7. We decided not to include them in the list because we did not detect any mutation. Thanks to the reviewer to bring this observation.

Other comments are followings:

Page 7, “Next, we performed whole-exome sequencing of the four ROC lines with excellent coverage mapping using the GRCm38 (mm10) C57BL/6J mouse genome as a reference (Supplementary Table 3).”

3) *“The authors should describe the method for whole-exome sequencing, coverage numbers, and mean coverage depth numbers.”*

We have included the information requested by the reviewer (**lines 138-142, p. 7-8; Supplementary Table 1**)

Page 8, “Furthermore, we assessed the association between the ROC cell line mutations and the most significantly mutated genes in The Cancer Genome Atlas (TCGA)-HNSCC cohort.

Bioinformatic studies detected acquired mutations in p53, Notch1, Csm3, Med12l, Kmt2d, Fat1, Fat3, Fat4, Akap9, and Myh6 in murine ROC cell lines, a reported previously in human HNSCC (Fig. 1D).” and Figure 1D;

4) *“The authors should add each variant allele frequency and depth.”*

The p53 gene deletion in the ROC2 cell line was obtained from genetically engineered mouse model in which the p53 gene is flanked by LoxP sites. The deletion was confirmed by using specific PCR primers as reported by Jokerson et al. (PMID: 11694875). The p53 T122N mutation in the ROC1 cell line was confirmed Sanger sequencing. We have included a **Supplementary table 4** that contains the sequencing details, including the variant allele frequency and depth of the mutated genes in the ROC cell lines (**lines 162 – 163, p. 8**).

5) *“Some genes are not defined as “cancer gene” by OncoKB database. Please revise the gene list.”*

The reviewer is correct that not all of the listed genes are defined as “cancer genes” in the OncoKB database. We chose to include more than just the OncoKB subset of genes because the TCGA lists included additional genes. We were also careful to not describe the gene list as “cancer genes” rather as frequently altered genes also found in TCGA.

6) *“Please describe how to calculate the mutation frequency of TCGA-HNSCC because the numbers are seemed to be incorrect.”*

We have updated the mutation frequency of TCGA-HNSCC table and included a more detailed information of the number frequency. (**Line 156-158, p. 8; Ref 26, 27**) the numbers are based on 285 HPV- HNSCC tumors from the oral cavity site. The data are from the TCGA PanCanAtlas PanSquamous project and cBioPortal.org. (PanSquamous Reference: Campbell et al. Cell Rep.

2018 Apr 3;23(1):194-212.e6. doi: 10.1016/j.celrep.2018.03.063. PMID: 29617660; cBioPortal Reference: Cerami et al. The cBio Cancer Genomics Portal: An Open Platform for Exploring Multidimensional Cancer Genomics Data. Cancer Discovery. May 2012 2; 401. PMID: 22588877, and Gao et al. Integrative analysis of complex cancer genomics and clinical profiles using the cBioPortal. Sci. Signal. 6, p11 (2013) PMID: 23550210).

7) *“Did ROC4 have Q61H mutation in Kras not Hras? Page 10,*

The mutation is in the Kras gene according to whole exome sequencing data (**Figure 1D**).

“As described above, ROC1 cells acquired a p53 mutation; such an alteration is detected in 85% of HNSCC patients, suggesting that p53 mutations influence the TIME.” 8) *“The authors described “85%” in this sentence, inconsistent to Figure 1D. Please describe the source of this frequency.”*

The reviewer makes an important point. We revised the original manuscript and the frequency of TP53 mutation ranges from 75 to 85% in non-HPV-associated HNSCC, we have edited accordingly (**Line 78, p 5; Lines 216-217, p. 11**). Our table shows 77% because the analysis only includes OSCC HPV-negative according to PanSquamous reference: Campbell et al. Cell Rep. 2018 Apr 3;23(1):194-212.e6. doi: 10.1016/j.celrep.2018.03.063. PMID: 29617660.

Page 11, “We next performed RNAseq of ROC1 tumors grown in immunocompetent C57BL/6Jmice to assess immune infiltration and identify tumor cell–intrinsic factors that might promote tumorigenesis and immunosuppression mechanisms.”

9) *“The authors should describe the method for RNA-seq and number of uniquely mapped reads. Page 11,*

We have included the information requested by the reviewer (**Lines 230-234, p. 12**) and incorporated a **Supplementary Table 5** containing detailed mapped reads.

“Bioinformatics analysis revealed that ROC1 tumors expressed markers of macrophages and Tregs as well as immune checkpoints associated with these cell types (Supplementary Fig. 5A).”
10) *What did the authors compare?*

Here we used the normalized RNA-seq of the ROC1 tumors that was compared to ROC1 cell line to identify the expression of specific immune cell markers and immune checkpoints. The two panels in **Supplemental Fig 5A-B**, were separated to have two different scale bars and observed the different level of expression of high and low immune genes. The RNA-seq data was compared with the ROC1 cell line.

11) *I'm interested in RNA-seq results in other ROC cells. Please mention why the authors didn't subject those cells for RNA-seq.*

We focus primarily on the ROC1 cell line to understand the tumor intrinsic factors and the mechanism of immunotherapy resistance. We did not want to diverge the focus of the manuscript with other data with no impact in the discussion of this manuscript. Yet, we are preparing a second manuscript, in which we are planning to include more studies of the other ROC cell lines as well more RNAseq data and HTG Edge-seq studies.

Page 11, *"We assessed the transcriptome analysis in ROC1 p53-KD over control cells and revealed a dramatic enrichment of proinflammatory cytokines and chemokines confirmed by gene set enrichment analysis (Fig. 4D)."*

12) *Fig. 4D seems like indicating the result of differentially expressing genes, not of gene set enrichment analysis.*

The reviewer is correct, we modify the term for differential expressing genes (**Line 250, p. 12**)

13) *"Please describe the method for this analysis. Page 12, "In ROC1 p53-KD cells, IL33 and ST2 receptor were both downregulated (Fig. 4E);"*

We have included the method as suggested by the reviewer (**Line 267, p. 13**)

the IL33/ST2 signaling pathway has been reported to sustain tumor-associated macrophages and activate tumor growth factor-beta (TGF- expression to sustain a signaling loop to promote cancer progression."

14) *"TGF- " is garbled."*

We have edited the error as indicated by the reviewer (**Line 269, p. 13; line 397, p.19**)

Page 13, *"We exposed BMCs and splenocytes to ROC1 control- and ROC p53-KD-conditioned media for 72 hours and identify the different immune cell populations through flow cytometry (Supplementary Fig. 6 and 7)."*

15) *"BMCs" should be replaced to "bone marrow cells (BMCs)".*

We have edited the sentence as requested (**Line 284, p. 14**)

Page 24, “Genomic DNA and RNA concentrations were determined using the NanoDrop system (Thermo Scientific, Wilmington, DE).”

16) In case the genomic DNA was subjected to whole-exome sequencing, dsDNA quantification using such as Qubit would be required. As for RNA-seq, the assessment of RNA integrity is a critical.

The samples were processed by BGI Genomics Company to perform the WES and RNA-seq studies. We have indicated in the Material and method section (**Line 531, p. 25**) that BGI Genomics Company generated the WES and RNA-seq data.

Page 25, “Reverse transcription-quantitative PCR (RT-qPCR). We used Bio-Rad CFX96 Touch Real-Time PCR Detection System and SsoAdvanced Universal SYBR Green Supermix (Bio Rad, USA) ... Each reaction was performed in triplicate.”

17) The authors should describe the method for reverse transcription reaction and internal control gene name (Gapdh).

We have included the information as indicated by the reviewer in the material and method section (**Line 554-557, and Line 562-563, p. 26-27**)

18) The number of references is out of order.

We have revised carefully the references as suggested. Thanks,

19) Page 5 of Supplementary Materials and Methods; “Western blotting” section was duplicate with “Methods”.

We have deleted the western blot method in Supplementary Materials and Methods section, to avoid duplicated information.

Reviewers' comments:

Reviewer #1 (Remarks to the Author):

Authors replied to all my comments. Few points should be furtherly addressed.

Extensive english sentence construction editing is needed. I find the manuscript difficult to read.

1)I would change this and

"177,757 patients died of these diseases" to "accounting for 177,757 disease-related deaths. or to "177,757 patients died of this disease" since the main disease is squamous cell carcinoma and usually statistics report together oral and lip cancer.

2) Probably mistyping here: " distantly become recurrent."

3) Line 99-104 change HNSCC to OSCC

4)Line 214 change HNSCC to OSCC and related statistics (I could not find a proper statistics for OSCC, please try to find it)

5) Line 216 change HNSCC to OSCC and related statistics:65-85% (Caponio, V.C.A., Troiano, G., Adipietro, I. et al. Computational analysis of TP53 mutational landscape unveils key prognostic signatures and distinct pathobiological pathways in head and neck squamous cell cancer. Br J Cancer 123, 1302–1314 (2020). <https://doi.org/10.1038/s41416-020-0984-6> AND Lindemann A, Takahashi H, Patel AA, Osman AA, Myers JN. Targeting the DNA Damage Response in OSCC with TP53 Mutations. J Dent Res. 2018;97(6):635-644. doi:10.1177/0022034518759068)

6) I would include the acronym for trp53

Please check the whole manuscript and change HNSCC to OSCC and related content.

Reviewer #2 (Remarks to the Author):

Thanks for making the required changes

Reviewer #3 (Remarks to the Author):

The manuscript has been revised well.

I think this manuscript will be acceptable after some corrections, described bellow, have been done.

1. The method for RNA-seq should be described in the "Materials and Methods" session, not in the "Results" section.

2. STAR or HISAT2 should be used for RNA-seq mapping instead of TopHat2.

3. Please add the description of command options for WES and RNA-seq analyses.

4. Title of Supplementary Table S4; "Variable allele frequency" should be replaced to "Variant ...", I think.

Point-by-point-reply

Reply to Reviewers

We thank the reviewers for a clear, positive and comprehensive evaluation of our work. We address their comments as follows:

Reviewer #1

Authors replied to all my comments. Few points should be furtherly addressed. Extensive english sentence construction editing is needed. I find the manuscript difficult to read.

We have re-edited the language of the manuscript.

1)I would change this and "177,757 patients died of these diseases" to "accounting for 177,757 disease-related deaths. or to "177,757 patients died of this disease" since the main disease is squamous cell carcinoma and usually statistics report together oral and lip cancer.

We have edited as suggested. **(Line 59, p.4)**

2) Probably mistyping here: " distantly become recurrent."

We have edited. **(Line 61, p.4)**

3) Line 99-104 change HNSCC to OSCC

We have edited. **(Line 101, 105, p.6)**

4)Line 214 change HNSCC to OSCC and related statistics (I could not find a proper statistics for OSCC, please try to find it)

We reviewed relevant papers and were not able to find data describing response to checkpoint inhibition in OSCC alone. To improve clarity, it now read: ‘Because only 10%-15% of patients with advanced-stage HNSCC (including OSCC) respond to checkpoint inhibition’. **(Line 215, p.11)**

5) Line 216 change HNSCC to OSCC and related statistics:65-85% (Caponio, V.C.A., Troiano, G., Adipietro, I. et al. Computational analysis of TP53 mutational landscape unveils key prognostic signatures and distinct pathobiological pathways in head and neck squamous cell cancer. Br J Cancer 123, 1302–1314 (2020). <https://doi.org/10.1038/s41416-020-0984-6> AND Lindemann A, Takahashi H, Patel AA, Osman AA, Myers JN. Targeting the DNA Damage Response in OSCC with TP53 Mutations. J Dent Res. 2018;97(6):635-644. doi:10.1177/0022034518759068)

We have edited and included the references suggested. (**Line 218, p.11**)

6) I would include the acronym for trp53

We had included the acronym for trp53 in abstract (**Line 44, p.3**), and included another acronym reference in main text body. (**Line 96, p.5**)

Please check the whole manuscript and change HNSCC to OSCC and related content.

We re-checked the manuscript and made corrections where needed. (**Line 70, p.4**) (**Line 378, p.18**)

Reviewer #3

The manuscript has been revised well. I think this manuscript will be acceptable after some corrections, described below, have been done.

1. The method for RNA-seq should be described in the "Materials and Methods" session, not in the "Results" section.

We have edited this part as suggested (**Line 531-535, p.25**)

2. STAR or HISAT2 should be used for RNA-seq mapping instead of TopHat2.

Our pipeline, at that time of the analysis used TopHat2 (2020). We are currently using STAR, however the results of the two pipelines should not make major difference. STAR is faster, but the results should be consistent.

3. Please add the description of command options for WES and RNA-seq analyses.

We have included the information requested in the Materials and Methods section (**Line 530 – 530, p25-26**)

4. Title of Supplementary Table S4; "Variable allele frequency" should be replaced to "Variant ...", I think.

We have edited it. (**Supplementary Table S4**)

REVIEWERS' COMMENTS:

Reviewer #3 (Remarks to the Author):

The authors responded to all of my concerns adequately.
I don't have any comment.